# Combinatorial optimization of gene expression through recombinase-mediated promoter and terminator shuffling in yeast

Charlotte Cautereels[1,2], Jolien Smets [1,2], Peter Bircham [1,2], Dries De Ruysscher[3,4], Anna Zimmermann [1,2], Peter De Rijk[5,6], Jan Steensels [1,2], Anton Gorkovskiy [1,2], Joleen Masschelein [3,4] & Kevin J. Verstrepen [1,2] ✉

Microbes are increasingly employed as cell factories to produce biomolecules. This often involves the expression of complex heterologous biosynthesis pathways in host strains. Achieving maximal product yields and avoiding build-up of (toxic) intermediates requires balanced expression of every pathway gene. However, despite progress in metabolic modeling, the optimization of gene expression still heavily relies on trial-and-error. Here, we report an approach for in vivo, multiplexed Gene Expression Modification by LoxPsym-Cre Recombination (GEMbLeR). GEMbLeR exploits orthogonal LoxPsym sites to independently shuffle promoter and terminator modules at distinct genomic loci. This approach facilitates creation of large strain libraries, in which expression of every pathway gene ranges over 120-fold and each strain harbors a unique expression profile. When applied to the biosynthetic pathway of astaxanthin, an industrially relevant antioxidant, a single round of GEMbLeR improved pathway flux and doubled production titers. Together, this shows that GEMbLeR allows rapid and efficient gene expression optimization in heterologous biosynthetic pathways, offering possibilities for enhancing the performance of microbial cell factories.

Microbial cell factories are increasingly used for the sustainable production of biofuels, bioplastics, food substitutes, medicines and other high-value compounds[1,2]. However, the successful implementation of heterologous pathways and perturbation of native microbial metabolism often requires extensive optimization to ensure economically viable production titers. Metabolic engineering allows rewiring of cellular metabolism to increase product titers by enhancing precursor supply[3,4], tackling co-factor limitations[5–7], interrupting competitive pathways[8,9], preventing buildup of toxic intermediates[10] and improving heterologous pathway flux[11]. Over the past years, several tools and strategies have been developed to facilitate yield optimization, including chassis strain improvement[12–14], directed mutagenesis[15–17], synthetic compartmentalization[18,19] and pathway gene expression optimization[20].

One of the most important yet challenging strategies for optimizing production yields is fine-tuning the expression levels of

[1]VIB Laboratory for Systems Biology, VIB-KU Leuven Center for Microbiology, Leuven 3001, Belgium. [2]Laboratory of Genetics and Genomics, Center of Microbial and Plant Genetics, Department M2S, KU Leuven, Gaston Geenslaan 1, Leuven 3001, Belgium. [3]Molecular Biotechnology of Plants and Micro-organisms, Department of Biology, KU Leuven, Kasteelpark Arenberg 31box 2438 Leuven 3001, Belgium. [4]Laboratory for Biomolecular Discovery & Engineering, VIB-KU Leuven Center for Microbiology, Leuven 3001, Belgium. [5]Neuromics Support Facility, VIB Center for Molecular Neurology, VIB, Antwerp 2610, Belgium. [6]Neuromics Support Facility, Department of Biomedical Sciences, University of Antwerp, Antwerp 2610, Belgium. ✉e-mail: kevin.verstrepen@kuleuven.be

individual genes in a biosynthetic pathway. Tuning gene expression levels allows to balance reaction flux, thereby maximizing product yields, while minimizing detriment to cell fitness, for example by maintaining native metabolic flux and reducing the burden of excessive protein synthesis[21–25]. Given the importance of tuning gene expression levels, it is not surprising that many studies choose to focus on the characterization of expression modulators, including promoters, terminators, ribosomal binding sites, transcription factors and untranslated regions (UTRs)[26–29]. In addition to natural elements, sets of artificial expression modulators have been developed[30–37]. Together, these provide a diverse molecular toolbox for precise modulation of gene expression.

While modification of expression and regulation of genes has become relatively easy, optimizing expression levels of a full metabolic pathway remains challenging. Various computational approaches for metabolic flux modeling have recently been exploited[38,39], but the development of these models relies on large, well-defined datasets, which are laborious and expensive to obtain[40]. Moreover, construction of kinetic models is hampered by the difficulty of acquiring exact kinetic reaction parameters due to uncertainties intrinsic to biological systems. These uncertainties need to be hypothesized or modeled based on approximates, which requires intense computational power and often renders researchers to oversimplification and unreliable predictions[41]. Therefore, the use of such in silico tools for targeted pathway optimization is currently still limited.

To experimentally improve pathway flux by generating large libraries of gene variants with different expression levels, several tools have been developed. For example, several papers describe the (ex vivo) parallel assembly of genes and expression modulators, such as (artificial) promoters[42,43], transcription factors[44], intergenic regions[45], and ribosomal binding sites[46], which can then be integrated in the microbial host. Other strategies often rely on CRISPR-based gene (in)activation systems that exploit gRNA oligo pools to target Cas proteins, either as functional editors[47], deactivated road blocks[48] or transcription factor fusion proteins[49], to specific loci for gene expression regulation. However, these strategies often require high technical skill and large, expensive oligonucleotide libraries, which strongly limits their applicability. Therefore, despite the large number of tools and techniques for gene expression diversification, none of these technologies provide cheap, fast, in vivo, multiplexed, large-range expression modification of multiple genes in parallel. In this study, we therefore aimed at developing a technique that would allow to quickly and easily generate a vast library of variants of a given starting strain, with each variant showing different expression levels and regulation of a set of selected genes. Our strategy is based on the use of the site-specific Cre-LoxP recombinase system.

Site-specific recombinases, that invert or excise DNA sequences flanked by specific target sites, are often used in synthetic biology circuits to control gene (in)activation and cellular behavior, e.g., by deleting a promoter upstream of a target gene[50–52]. Among these recombinases, Cre recombinase is widely used due to its efficiency in many organisms and its independence from any accessory proteins[53–55]. Cre recombinase recognizes a short 34 bp DNA sequence (LoxP), binds it as a dimer and forms a tetrameric complex with another LoxP-bound dimer to recombine the two target sequences and enforce DNA cutting and pasting[56]. Depending on the orientation of the LoxP sequence, the flanking DNA will either be inverted or deleted. To bypass this directionality, a symmetrical LoxPsym site was developed, thereby expanding the types of structural variation that can be achieved upon recombination[57]. LoxPsym is a key element in the *Saccharomyces cerevisiae* 2.0 project, a synthetic biology project that aims to synthesize the first eukaryotic genome[58,59]. In *Sc* 2.0, LoxPsym sites are inserted across the genome after the ORF of each non-essential gene to enable massive genome shuffling and rapidly generate phenotypic diversity, a technology referred to as SCRaMbLE

(Synthetic Chromosome Rearrangement and Modification by LoxP-mediated Evolution). Several applications of SCRaMbLE for the improvement of industrial production traits have been reported in recent years, demonstrating the potential and importance of structural variation for strain optimization[60–63].

In this study, we develop an approach called GEMbLeR (Gene Expression Modification by LoxPsym-Cre Recombination) that exploits site-specific recombination[63] for rapid and in vivo gene expression diversification in yeast. We first characterize a pool of hybrid promoters and terminators flanked by LoxPsym sites and use a minimal yet diverse selection to design a hyper-evolvable Gene Expression Modulator (GEM) construct. To demonstrate the capabilities of GEMbLeR, we use a fluorescent reporter system to generate variants with protein expression spanning over two orders of magnitude. Finally, we apply GEMbLeR for multiplexed, combinatorial expression optimization of six heterologous genes of the astaxanthin biosynthesis pathway and show an improvement in production titers of more than two-fold. Together, this shows that GEMbLeR is an efficient, fast and inexpensive technology for gene expression optimization of metabolic pathways.

## Results

### Design of a hyper-evolvable gene expression modulator (GEM)

To develop a tool for in vivo gene expression diversification in *S. cerevisiae*, we designed sets of recombinable promoter and terminator sequences (GEM-blocks), which we assembled into arrays for the construction of a 5′ and 3′ GEM module (Fig. 1a). The 5′ GEM consists of an array of upstream promoter elements (UPEs) separated by LoxPsym recombination sites, whereas the 3′ GEM contains a set of different terminator sequences separated by LoxPsym sites. The sets of UPEs and terminators used in the arrays vary greatly in strength and gene expression regulation. The GEM system can drive the expression of a target gene by replacing its native promoter and terminator with the 5′ and 3′ GEM arrays, respectively. As recombination of LoxPsym sites can result in deletion, inversion, translocation and duplication of each GEM-block, induction of Cre recombinase would lead to a virtually unlimited pool of GEM variants, that each have a different effect on gene expression, ultimately resulting in a pool of phenotypic diversity within the population (Fig. 1b). Moreover, using different orthogonal LoxPsym recombination sites that do not cross-react[63] in each GEM module prevents recombination between the 5′ and 3′ GEM, as well as between GEMs linked to other genes. As sixteen of such orthogonal sites have been identified[63], 5′ and 3′ GEMs can be added to a maximum of eight genes, enabling the use of the tool in full pathways.

One of the requirements of our tool is the introduction of LoxPsym sites at the promoter and terminator regions, which can influence expression regulation of the targeted gene. Therefore, we tested the effect of inserting LoxPsym sequences at different positions in the GEM-blocks, using *yECitrine* as a reporter gene. For the 5′ GEM, we evaluated the introduction of a LoxPsym site at three positions: upstream of the TATA box, upstream of the transcription start site (TSS), or upstream of the start codon (ATG). These positions were chosen strategically to evaluate whether the LoxPsym sequence could influence recruitment and assembly of the transcription pre-initiation complex, transcription initiation and translation initiation, respectively. The results revealed that insertion of LoxPsym at the promoter region reduces protein levels, with increasing repression when LoxPsym is moved closer to the open reading frame (ORF) (Fig. 1c). Notably, insertion of LoxPsym directly in front of the start codon reduced protein production to a level comparable to that of the non-fluorescent control strain. To test if these observations were promoter-specific, we constructed the same promoter edits for the weak *PGI1* and the intermediate *TPI1* promoter which also showed a similar trend (Supplemental Fig. 1a–d, Supplementary Data 1, promoter

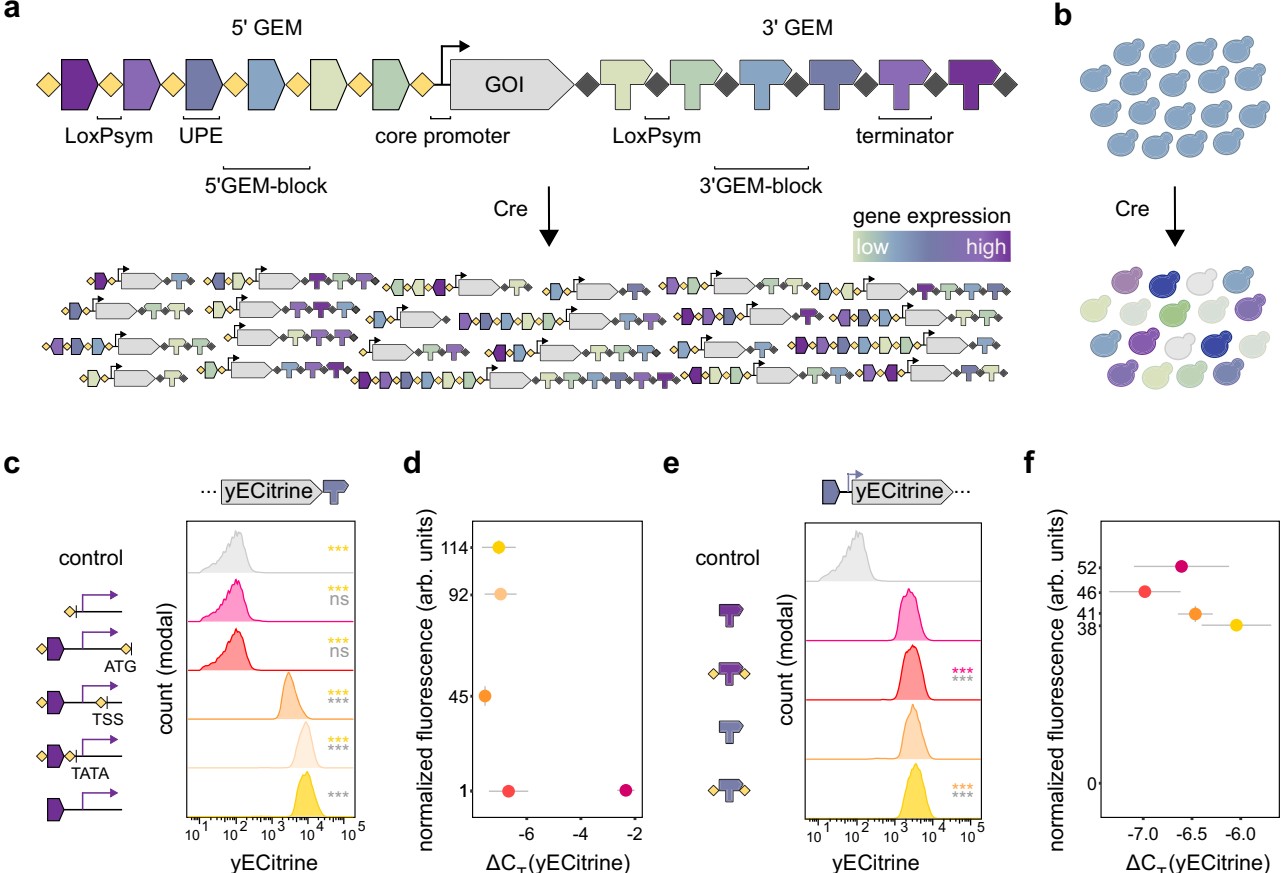

**Fig. 1 | Shuffling upstream promoter elements (UPEs) and terminators by Cre-LoxPsym recombination for in vivo expression diversification. a** The gene expression modulator (GEM) design. 5' and 3' GEM consist of an array of UPEs and terminators, respectively, separated by LoxPsym sites. Different promoter and terminator parts have different strengths (color scale). Upon induction of Cre recombination, blocks in each GEM are shuffled, resulting in a pool of GEM variants that differently influence expression of the gene of interest (GOI). Different, orthogonal LoxPsym sites restrict recombination to one GEM module. **b** Inducing recombination of GEMs generates a cell population with diversified GOI expression levels (different colors). **c** Effect of introducing LoxPsym (5′-ATAACTTCGTA-TATTATATAATATACGAAGTTAT-3′) at the *TDH3* promoter on yECitrine fluorescence. LoxPsym was placed either directly upstream of the start codon (ATG), transcription start site (TSS) or TATA box, each time in combination with a LoxPsym site upstream of the promoter region. The strength of the native promoter (yellow) and LoxPsym-following core promoter (pink) was also measured. Constructs were tested in combination with terminator *CYC1* and integrated at the *CAN1*

locus of the laboratory strain *BY4741-mCherry* (the control strain, gray). Histograms represent concatenated populations of three biological repeats. Statistics by analysis of variance and two-sided Tukey multiple comparisons of means (exact *p*-values in Supplementary Data 1, *p* values LoxPsym position). Colors of *p* values indicate the population used for comparison (***$p < 0.001$, 'ns' $p > 0.1$). **d** yECitrine fluorescence versus mRNA abundance, measured via qPCR ($\Delta C_T$ value). Dots represent average of three biological replicates, colors correspond to (**c**). Horizontal and vertical error bars represent standard error and standard deviation, respectively. **e** Effect of flanking a yeast terminator (*HIS5* (purple) and *CYC1* (blue)) with LoxPsym on yECitrine fluorescence. Constructs were tested in combination with promoter *TPI1* and integrated at the *CAN1* locus of the laboratory strain *BY4741-mCherry*. Histograms show concatenated data of three biological repeats. Statistics similar to (**c**). **f** Normalized fluorescence versus mRNA abundance of yECitrine, colors correspond to (**e**), dots and error bars similar as (**c**). Source data for this figure are provided as a Source Data file.

sequences). Interestingly, transcript analysis by qPCR revealed that the observed decrease in protein levels was not caused by a reduction of transcription (Pearson correlation test with $p = 0.3326$), indicating that inserting the LoxPsym site in the 5′ UTR likely inhibits translation (Fig. 1d), possibly through the formation of a hairpin structure in the mRNA formed by hydrogen bonds between the palindromic LoxPsym arms and part of the symmetrical spacer[64,65] (Supplemental Fig. 1e) or disruption of the Kozak sequence[66–68] (Supplemental Fig. 1h). To minimize the negative effect of the LoxPsym site on gene expression at these positions, we attempted to construct a recombination site with a weaker secondary structure. To do this, we modified one of the LoxPsym palindromic arms using modifications that can be recognized efficiently by a slightly adapted recombinase[69] (Supplemental Fig. 1f–g). However, these modifications did not restore expression levels. We therefore decided to introduce the LoxPsym site in front of the TATA box to minimize 5′ GEM construct size while minimally

impacting translation. The region from the TATA box to the start codon is further referred to as the 'core promoter', which will be fused to different UPEs upon recombination.

The design of the 3′ GEM aimed to enable shuffling of terminator elements. Since insertion of the LoxPsym site 3 bp downstream of the ORF has a minor effect on transcript level[58,70], we used the region starting directly behind the stop codon until the end of the terminator (as defined in previous literature (Supplementary Data 1, terminators sequences)), flanked by loxPsym sites on both sides, as 3′ GEM blocks. We tested the effect of this design on an intermediate (*CYC1*) and a strong (*HIS5*) terminator (Fig. 1e, f). We observed a minor decrease in protein levels for both terminators, but, similar to the 5′ GEM, we were unable to directly correlate this with transcript levels (Pearson correlation test with $p = 0.3540$). Since inserting LoxPsym in the 3′ UTR did not show major expression limitations, we continued with this design.

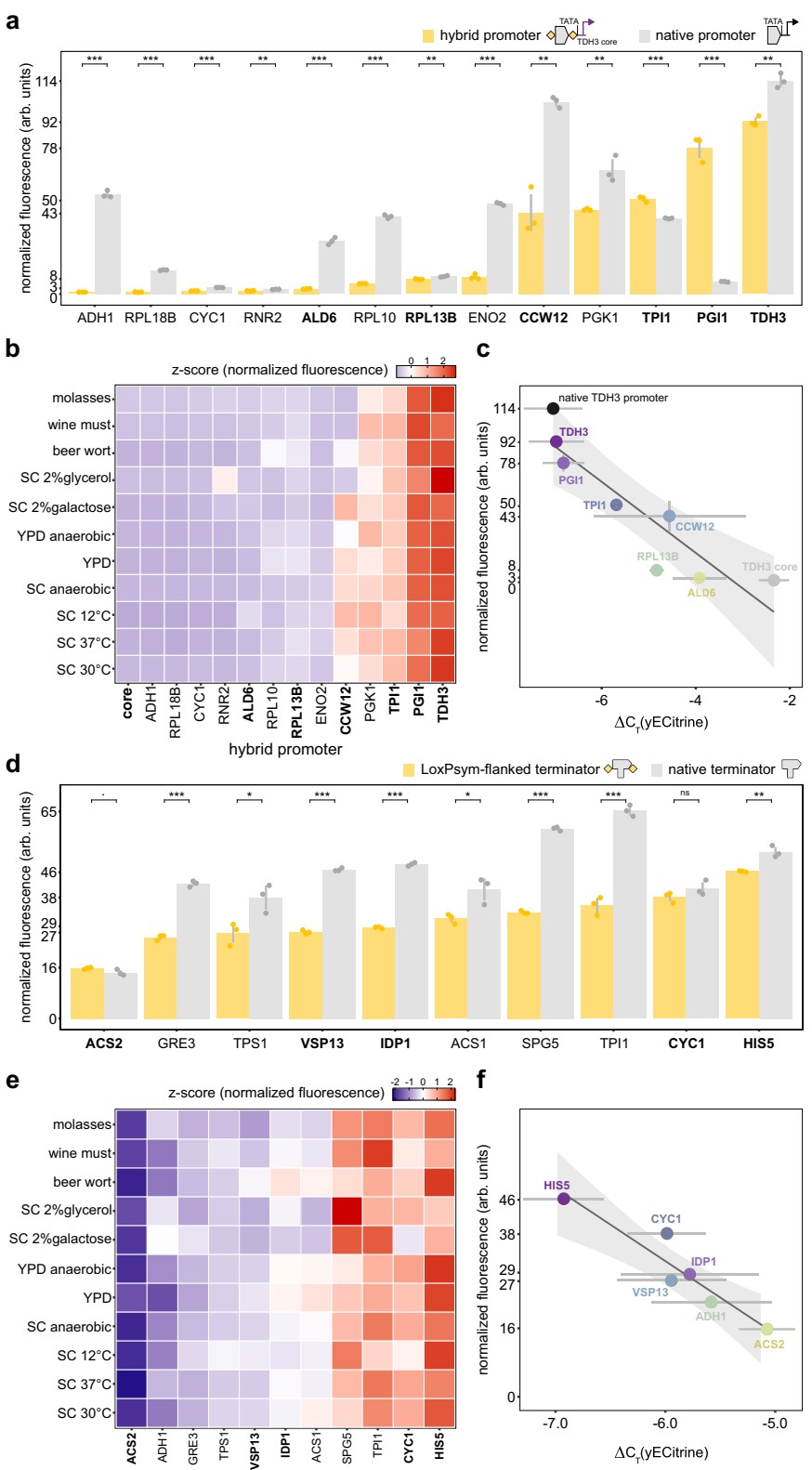

## GEM blocks span a wide expression range in divers conditions

To ensure that recombination of GEM would lead to a suitable range of expression levels, we compared the expression levels obtained with various potential GEM-blocks to those of commonly used promoters and terminators for heterologous gene expression[71–74] (Supplementary Data 1, promoter and terminator sequences). For the 5′ GEM blocks, we limited our selection on UPEs of TATA-containing promoters (Fig. 2a),

as eukaryotic TATA-promoters are typically associated with enhanced transcription and translation efficiency[75]. To minimize the length of the UPEs, and thereby reduce the GEM DNA synthesis cost, we first tested the effect of reducing the promoter region to 500 bp for three *S. cerevisiae* promoters: a weak (*PGI1*), intermediate (*TPI1*) and strong (*TDH3*) promoter. The results showed that expression levels obtained with the truncated promoters were comparable to those obtained with

**Fig. 2 | Selection of promoter and terminator GEM-blocks with diverse activity.** **a** yECitrine fluorescence regulated by 13 hybrid promoters (*TDH3* core promoter preceded by a LoxPsym-flanked UPE, yellow), in combination with terminator *CYC1*. Gray bars indicate natural promoters of the corresponding UPEs. Selected 5′ GEM-blocks are indicated in bold. Bars and error bars represent average and standard deviation of three biological replicates, respectively. Unpaired two-sample two-sided *t* tests to compare the hybrid and native promoters, with $p = 1.21e\text{-}06$ (*ADH1*), $1.29e\text{-}07$ (*RPL18B*), $4.89e\text{-}06$ (*CYC1*), $0.001845$ (*RNR2*), $2.07e\text{-}05$ (*ALD6*), $3.69e\text{-}07$ (*RPL10*), $0.003511$ (*RPL13B*), $1.76e\text{-}06$ (*ENO2*), $0.001156$ (*CCW12*), $0.006358$ (*PGK1*), $0.0002717$ (*TPI1*), $5.44e\text{-}05$ (*PGI1*) and $0.001312$ (*TDH3*). **b** Activity of hybrid promoters in 11 different media (y-axis), represented by z-score of normalized fluorescence levels of three biological replicates, calculated by medium. **c** Pearson correlation test between yECitrine fluorescence and mRNA abundance ($\Delta C_T$ value) of selected 5′ GEM-blocks ($R^2 = 0.82$, $p = 0.0021$). Dots, horizontal and vertical error bars represent average, standard error and standard deviation of three biological replicates, respectively. Colors (shades of purple, blue and green) correspond to

the UPE element used in the hybrid promoter with the *TDH3* core. Gray represents the *TDH3* core promoter by itself and black the native *TDH3* promoter. **d** yECitrine fluorescence regulated by the *TPI1* promoter and 10 terminator sequences, either flanked by LoxPsym sites (yellow) or not (gray). Selected 3′ GEM-blocks are indicated in bold. Bars and error bars represent average and standard deviation of three biological replicates, respectively. Unpaired two-sample two-sided *t* tests to compare LoxPsym-flanked and natural terminator sequences with $p = 0.07434$ (*ACS2*), $2.23e\text{-}05$ (*GRE3*), $0.0271$ (*TPS1*), $1.80e\text{-}06$ (*VPS13*), $5.86e\text{-}07$ (*IDP1*), $0.02329$ (*ACS1*), $4.91e\text{-}07$ (*SPG5*), $0.000103$ (*TPI1*), $0.176$ (*CYC1*), $0.004445$ (*HIS5*). **e** LoxPsym-flanked terminator activities in 11 different media (y-axis), represented by z-scores of normalized fluorescence levels of three biological replicates, calculated by medium. **f** yECitrine fluorescence versus mRNA abundance of selected 3′ GEM-blocks (Pearson correlation test with $R^2 = 0.89$, $p = 0.0046$). Dots, horizontal and vertical error bars represent average, standard error and standard deviation of three biological replicates, respectively. Color corresponds to LoxPsym-flanked terminator. Source data are provided as a Source Data file.

full-length promoters (Supplemental Fig. 2a). We therefore limited the region of the UPEs starting 500 bp upstream of the respective ORF and ending directly upstream of the TATA box.

To identify the most suitable core promoter, we combined the UPEs of *PGI1*, *TPI1* and *TDH3* with the core promoter of the weak (*PGI1*) and strong (*TDH3*) promoter (Supplemental Fig. 2b). Both promoters have been classified as Mediator-tail dependent promoters[76]. Mediator is a large (up to 21 subunits in yeast), variable protein complex that globally regulates the RNA polymerase II and is a general target of TF activation, which enhances stable interactions with RNA Polymerase II[77]. The activity of Mediator-tail dependent promoters depends on the interaction between the tail domain of the Mediator complex and transcription activators that are recruited to DNA binding sites upstream of the core promoter region[76]. We reasoned that the core sequences of these promoters would serve as effective substrates for gene expression regulation via UPE diversification. Our results revealed that the capacity of the UPEs to influence gene expression largely depended on the core promoter sequence, with the strong *TDH3* core supporting a broader range of expression levels compared to the core of the weaker *PGI1* promoter (Supplemental Fig. 2b). As such, we decided to use the *TDH3* core as the basis for the 5′ GEM design. Next, we combined the *TDH3* core with different UPEs, predominantly from other Mediator-tail dependent promoters, as combining UPEs and core promoters from the same class is generally important for optimal expression[78]. We also included *RPL18B*, *CYC1*, *RPL10*, *RPL13B* as Mediator-tail independent UPEs[76] and observed that all of these UPE$_X$-core$_{TDH3}$ hybrids induced only weak expression levels (Fig. 2a).

Interestingly, the correlation between the hybrid and natural promoters was poor (Pearson correlation test with $p = 0.05838$), indicating that the expression levels are likely influenced by complex interactions between the UPE and core promoter (Supplemental Fig. 2c). The majority of the hybrid promoters (11/13) showed reduced protein levels compared to their natural counterpart, with the strongest reduction showing a 55.6-fold decrease in fluorescence. To ensure our tool can be used in a wide array of applications, e.g., to optimize pathways in different fermentation settings, we tested these hybrid promoters in various industrially relevant environments (different carbon sources, temperature stress, oxygen stress and industrial fermentation media) (Fig. 2b). Construction of a linear mixed-effects model revealed that the interaction between the promoter sequence and the environmental condition was significant, indicating that the hybrid promoter is influenced by the environment (restricted likelihood ratio test with $p < 2.2e\text{-}16$, Supplemental Table 1). Nevertheless, we observed that general trends of expression regulation amongst the hybrid promoters are maintained across different conditions (Fig. 2b).

Based on the obtained data, we selected six LoxPsym-flanked UPEs as 5′ GEM-blocks, showing a high expression range that

resembled the natural promoter activity in *S. cerevisiae* (Supplemental Fig. 2d, e). Analysis of the transcript levels revealed a strong correlation between the protein levels (as assayed through fluorescence measurements) and the mRNA abundance ($R^2 = 0.82$, Pearson correlation test with $p = 0.002125$, Fig. 2c). Notably, the *TDH3* core promoter itself resulted in very limited expression, indicating that the UPEs are driving stronger transcription.

Similar to our strategy for the 5′ GEM, we tested several terminator sequences as potential candidates for incorporation into the 3′ GEM (Fig. 2d). Flanking these terminator regions with LoxPsym sites generally did not change relative terminator strength ($R^2 = 0.47$, Pearson correlation test with $p = 0.02928$, Supplemental Fig. 3a). To ensure that the same level of expression alteration could be achieved when modifying the promoter strength, the LoxPsym-flanked terminators were tested in combination with both the intermediate *TPI1* promoter as well as the stronger *TDH3* promoter (Supplemental Fig. 3b). The data revealed that different terminators can modulate expression levels over a range of approximately three-fold for both promoters, with consistent trends observed for both ($R^2 = 0.61$, Pearson correlation test with $p = 0.01308$). Next, we defined a set of LoxPsym-flanked terminators that maximized the expression range for incorporation in the design of the GEM (Supplemental Fig. 3c). In contrast to the 5′ GEM-blocks, their effect on gene expression was not significantly affected by the environment (restricted likelihood ratio test with $p = 0.1238$, Fig. 2e, Supplemental Table 2). Finally, we observed that transcript and fluorescence levels correlated well ($R^2 = 0.89$, Pearson correlation test with $p = 0.004613$, Fig. 2f).

## GEMbLeR for fast, wide range expression optimization in vivo

Following the selection of suitable GEM-blocks, we finalized the layout of the 5′ and 3′ GEM (Fig. 3, Supplemental Figs. 2e, 3d). We positioned weaker genetic elements (both for the UPE and terminator blocks) closer to the ORF to avoid that strong elements close to the ORF(s) might push the production of heterologous protein(s) to levels that are toxic for the initial, pre-recombination host strain. To assess the potential of GEM for rapid, in vivo expression diversification, we measured yECitrine fluorescence for three different setups: using the 5′ GEM, the 3′ GEM or both simultaneously (GEM). Additionally, cells were equipped with an inducible *Cre* recombinase expression plasmid or an empty vector (control). To determine the optimal induction time for the Cre recombinase, we tested various induction periods with cells bearing the 5′ GEM and observed that expression variation was largest for an induction period of 6 h (Supplemental Fig. 4a, b). We applied these conditions to all three setups, and the resulting populations showed significant differences in protein level variation between the recombined populations and the non-recombined controls (Fig. 3a–c).

Overall, we observed that diversification of the 5′ GEM (UPEs) resulted in a higher variation in fluorescence compared to the 3′ GEM

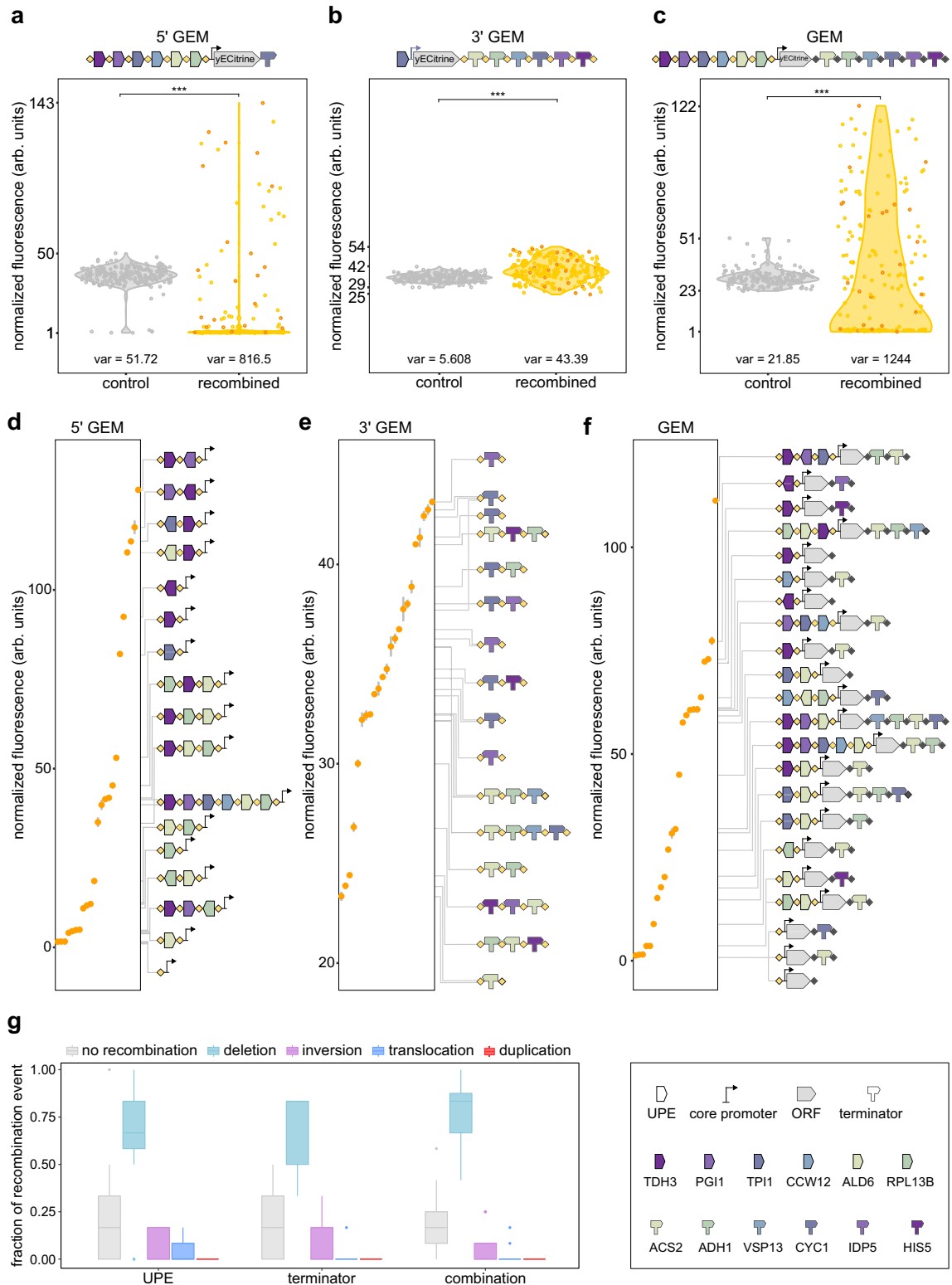

(terminators), consistent with the data obtained for the individual blocks. Surprisingly, 41 % of the diversified 5' GEM population did not show detectable yECitrine fluorescence (< the non-fluorescent control signal), indicating complete deletion of all 5' GEM-blocks and loss of heterologous protein production (Fig. 3a). Complete loss of fluorescence was not observed when only the 3' GEM was recombined (Fig. 3b). Importantly, this deletion bias was reduced to 7.2 %, when

combining the 5' and 3' GEM recombination (Fig. 3c). We have previously shown that multiplexed usage of LoxPsym variants in a sequential array can reduce the recombination efficiencies of LoxPsym sites positioned in the middle of the array[63]. Simultaneous recombination of 5' and 3' GEM might therefore reduce the recombination activity for the LoxPsym sites closest to the ORF, thereby reducing complete loss of all 5' GEM-blocks. Additionally, 3' GEM shuffling was

**Fig. 3 | Recombination effectively induces large-range gene expression variation of a fluorescent reporter.** Comparison of normalized fluorescence between a control group (gray) and a *Cre*-expressing group (yellow), carrying (**a**). the 5′ GEM (*N* = 210) in combination with *CYC1* terminator, (**b**). the 3′ GEM (*N* = 225) in combination with *TPI1* promoter or (**c**). a combination of the 5′ and 3′ GEM constructs (*N* = 180). The legend of the GEM modules is shown at the bottom right of the figure. Dots represent normalized fluorescence of separately induced clones randomly selected from six independently induced populations after plating. Statistics by Fligner-Killeen test with *p* = 7.20e-07 (UPE), 2.20e-16 (terminator) and 2.20e-16 (combination). Orange dots indicate clones which were further analyzed and sequenced (23 per construct), shown in (**d**). 5′ GEM, (**e**). 3′ GEM, and (**f**).

Combination of 5′ and 3′ GEM. Dots here represent the average of 3 biological repeats, error bars show the standard deviation. The sequences of the recombined GEM layouts are depicted on the right of each graph, connected to the corresponding dots with gray lines. **g** The frequency of no recombination, deletions, inversions, translocation events or duplications as calculated from the data shown in (**d–f**), obtained from sequences of single colonies after induction. The type of recombination event which occurred for each LoxPsym-flanked element (GEM-block) was counted for each setup and the distribution of fractions is shown by boxplots. The center line, box limits, dots and whiskers of the boxplots indicate the median, first and third quartiles, outliers and 1.5 x interquartile range, respectively. Source data for this figure are provided as a Source Data file.

tested in combination with the strong *TDH3* promoter, and the combination of 5′ and 3′ GEM was tested using a different combination of two orthogonal LoxPsym variants, of which one was reported to have a decreased activity when combined with other LoxPsym sites[63] (Supplemental Fig. 4c, d). Both variations allowed effective fine-tuning, although the use of a less active LoxPsym variant led to a reduced variance compared to that observed in Fig. 3c. We also noticed that the distribution of the diversified population centered around the initial expression levels, suggesting enhanced shuffling of 3′ GEM-blocks compared to the 5′ GEM-blocks, for which a less active LoxPsym variant was used. This indicates that the capacity of GEM shuffling depends on the LoxPsym variant[63].

To confirm that the observed changes in fluorescence were stable and caused by GEM recombination, we selected a range of variants from all three setups, measured the fluorescence for multiple biological repeats and identified their GEM layout by Sanger sequencing (Fig. 3d–f, Supplemental Fig. 5). Generally, we observed that GEM-blocks associated with high or low expression levels in the previous assay were enriched in high and low fluorescent clones, respectively. Importantly, diversification of 5′ GEM alone or in combination with 3′ GEM yielded multiple variants with higher expression levels than those obtained with the strongest building blocks individually. Moreover, for combined 5′ and 3′ GEM shuffling, we identified several clones with the same 5′ GEM layout that showed expression fine-tuning via the 3′ GEM (Fig. 3f), indicating that the combination of up- and downstream GEMs allows a more precise expression fine-tuning. Interestingly, inversions of the building blocks generally resulted in minor changes in fluorescence, with the inverted orientation sometimes leading to enhanced expression, confirming that the orientation of promoter elements can influence gene expression[79]. Surprisingly, the sequencing data also revealed that expression levels were not only influenced by the 3′ GEM-block that was closest to the ORF. Instead, the complete buildup of the 3′ GEM (terminator array) influenced expression. For example, clones for which the *yECitrine* reporter was followed directly by the *ACS2* terminator, with different downstream terminators present in their respective 3′ GEM, showed different fluorescence levels (Fig. 3e).

With regard to the types of the recombination events occurring, we observed a similar pattern for all three tested layouts (Fig. 3g). Overall, deletions were the most common recombination outcome (0.72 ± 0.058), followed by no recombination (0.20 ± 0.039), inversions (0.056 ± 0.022) and translocations (0.024 ± 0.018). Notably, we did not observe any duplications. Predominance of Cre-induced deletions, despite the symmetrical nature of the recombination site, has been observed previously for the synthetic chromosome arm synIXR. However, the absence of duplications is in contrast with previous findings[80].

Together, our findings indicate that although 3′ GEM shuffling by itself only generates limited variation in expression dynamics, combining 5′ and 3′ GEM shuffling yields superior results in terms of generating expression diversity and avoiding complete gene inactivity in a large fraction of the population. Sequence analysis of the recombined variants provided further confirmation that the combination of 5′ and

3′ GEMs increases diversity and decreases the number of identical clones in the shuffled populations.

## Multiplexed GEMbLeR improves astaxanthin production in yeast

To demonstrate the effectiveness of GEMbLeR for pathway flux optimization, we applied this tool to boost astaxanthin production in *S. cerevisiae*. Astaxanthin is an orange-red colored xanthophyll carotenoid beneficial for human health due to its anti-oxidant, anti-inflammatory and anti-cancer activity[81]. To produce astaxanthin, we introduced six heterologous genes (*tHMG1*, *CrtE*, *CrtI*, *CrtYB*, *CrtW*, *CrtZ*) under control of 5′ and 3′ GEMs at different genomic locations in the *S. cerevisiae* strain *BY4741*[55,82] (Supplementary Data 1, genes). We designated this starting strain 'ySTART'. We also constructed a reference strain in which the pathway genes were regulated by native yeast promoters and terminators commonly used for metabolic engineering purposes (Supplemental Table 3), referred to as 'yREF', thereby mimicking the conventional approach of pathway engineering. Both strains showed comparable growth rates, slightly slower than the original strain, indicating a limited cell burden caused by initial expression of the six heterologous genes under the control of GEMs (Supplementary Fig. 6a).

Before subjecting these strains to GEMbLeR, we assessed whether multiplexing imposed negative side effects that could compromise efficiency of the tool. First, we assessed whether implementation of multiple GEMs caused genomic instability. To this end, we grew the ySTART strain for over 100 generations and analyzed the lengths of the GEM regions up- and downstream of the six targeted pathway genes in eight randomly selected clones (Supplementary Fig. 6b–d). We observed genomic alteration in only three of the 96 fragments, indicating that the strains remained reasonably stable. Second, we evaluated whether expression variation capacity of GEMbLeR would be influenced by multiplexed usage of several identical GEMs in one genome, for example due to transcription factor depletion caused by repetitive usage of UPEs. The GEM-controlled *yECitrine* cassette used previously was introduced into the astaxanthin production strain and the expression variation profile following recombination was compared to that of a *BY4741* strain carrying only the GEM-*yECitrine* construct (Supplementary Fig. 7a). The results showed that variation in fluorescence could already be obtained after 2 h of *Cre* expression induction and that the variation was larger in the presence of other GEMs (Fligner-Killeen test with *p* = 2.896e-12), indicating that multiplexed usage of GEMs does not compromise the functionality of GEMbLeR.

We next induced Cre-mediated recombination in the ySTART strain for different induction periods and quantified the color (Red-Green-Blue, RGB, analysis) of randomly selected clones as a proxy for the production of different concentrations of astaxanthin and/or intermediate metabolites, such as lycopene, β-carotene, zeaxanthin, pheonicoxanthin and canthaxanthin[83] (Fig. 4b, c, Supplementary Fig. 7b–d). Importantly, the fraction of the induced population with a (near-) white color (high GB-values) was small (3%), indicating that only a few recombined clones had lost the phenotype of interest.

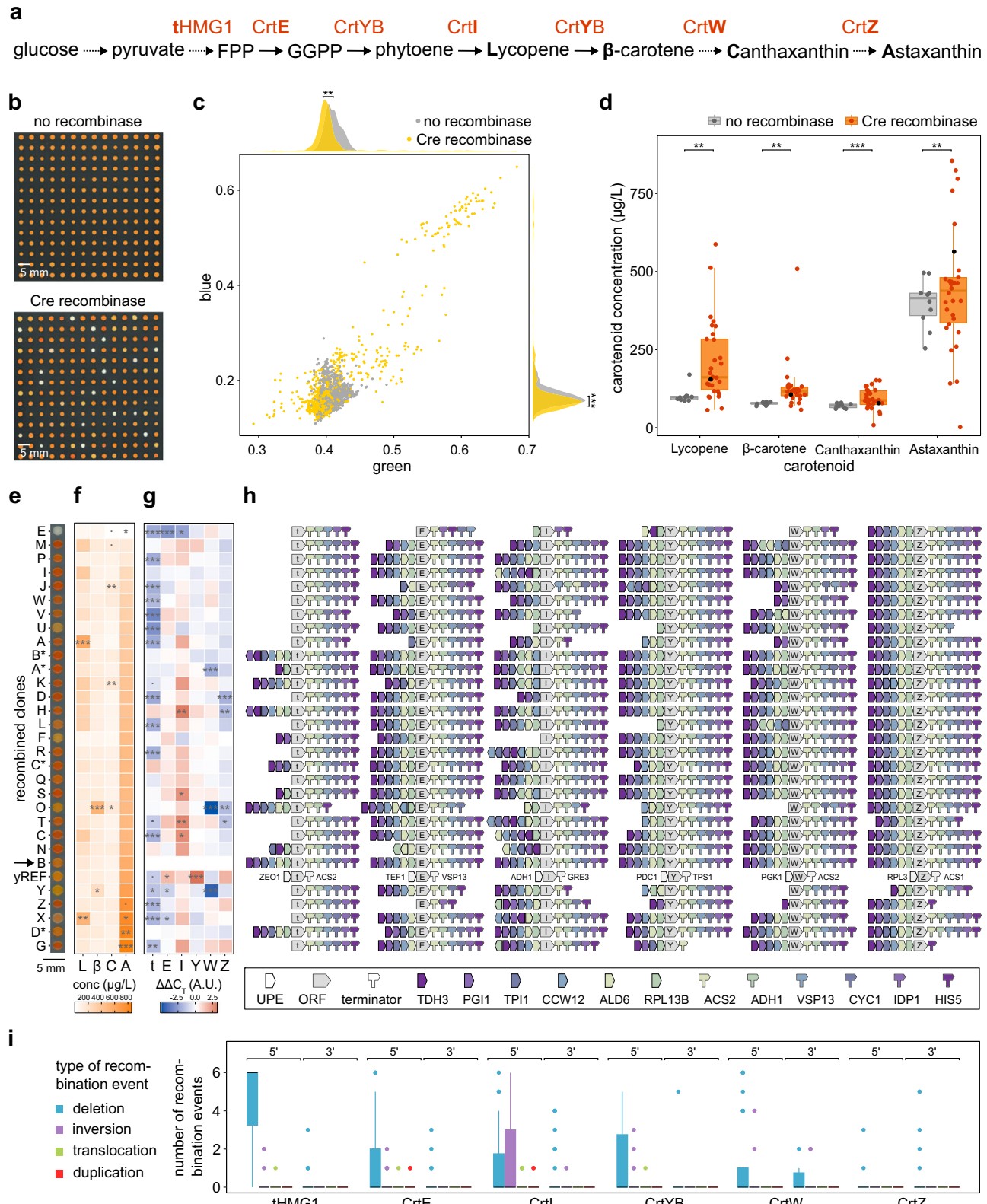

Moreover, no white-colored clones were observed in the control group, further confirming that spontaneous homologous recombination between different GEM constructs only rarely occurs. The average GB-values of induced clones were significantly lower compared to the control group, suggesting an overall increase in carotenoid production (Fig. 4c). To further investigate, we selected 30 diverse recombined strains based on their color, with a bias towards more red colored clones, as red colonies are generally associated with increased astaxanthin (or the red pathway intermediates lycopene and cantaxanthin) production[83] (Supplemental Fig. 7e). As expected, metabolite analysis revealed that the variation in carotenoid levels was significantly larger for the recombined clones compared to the ten control strains, with increased levels of red colored compounds (Fig. 4d). Importantly, we could not observe a

**Fig. 4 | GEMbLeR optimizes astaxanthin production in yeast. a** Astaxanthin biosynthesis pathway in *BY4741*. Heterologous genes targeted for expression optimization in orange (Supplementary Data 1, genes). Bold characters represent abbreviations of genes/compounds used in following panels. Full/dashed arrows indicate one/multiple enzymatic conversion(s). Conversion from β-carotene to astaxanthin goes through different routes (Supplemental Fig. 7f). **b** Single clones after GEMbLeR induction, carrying *Cre* expression plasmid (bottom) or empty vector (top). **c** GB-values of 1408 and 1490 single clones from induced populations with (yellow) and without (gray) Cre, respectively. Variances in GB-values were 0.0001566 and 0.0004302 (control), and 0.002291 and 0.007560 (recombinase). Statistics shown by Fligner-Killeen test with $p = 3.23e-03$ (x-axis) and 8.30e-10 (y-axis). Statistics by unpaired two-sided Wilcoxon test with $p = 2.40e-69$ (x-axis) and 8.40e-14 (y-axis). **d** Intracellular carotenoid concentrations for induced clones of recombinase (orange) and control (gray) groups. Dots represent average of three biological repeats (Supplementary Data 1, carotenoids). Clone B (identical to ySTART) in black. Statistics by Fligner-Killeen with $p = 0.002252$ (Lycopene),

0.006982 (β-carotene) and by two-sided F-test with $p = 5.765e-05$ (Canthaxanthin) and 0.006033 (Astaxanthin). Center line, box limits, dots and whiskers of boxplots indicate median, first and third quartiles, outliers and 1.5 x interquartile range, respectively. **e** 30 selected GEMbLeR variants, labeled alphabetically A-Z, A*-D*. Strain yREF is also shown. **f** Heat map with carotenoid concentrations per GEMbLeR variant (average of three biological repeats). Statistics by analysis of variance and two-sided Dunnettx's multiple comparisons of means (‘***’ $p < 0.001$, ‘**’ $p < 0.01$, ‘*’ $p < 0.05$, ‘.’ $p < 0.1$, Supplementary Data 1, stats carotenoids). **g** Gene expression data ($\Delta\Delta C_T$ value, qPCR of three biological repeats measured in technical duplicate, Supplementary Data 1, expression). Blue and red colors indicate an expression decrease and increase compared to clone B (no structural variation), respectively. Statistics similar to (**f**) (Supplementary Data 1, stats expression). **h** Structural variation after GEMbLeR. LoxPsym sites and *TDH3* core promoter not depicted. **i** Type of recombination event per locus (sequencing of single colonies after induction). Boxplots similar to (**d**). Source data are provided as a Source Data file.

clear relationship between color intensity and carotenoid levels by eye (Fig. 4e, f) and found that color variation only correlated with canthaxanthin concentrations (Pearson correlation test with $p = 0.00776$ (green) and 0.0100 (blue), Supplemental Fig. 8a, Supplementary Data 1, pairwise correlations). Nevertheless, we identified several improved clones (X, D* and G) that produced more than two-fold higher astaxanthin levels than the control group, with the best producing strain (G) showing a 2.15-fold improvement in production titer (Dunnettx multiple comparison of means with $p = 0.0007520$). Moreover, strain G also showed higher selectivity of astaxanthin production, with a astaxanthin/total carotenoid ratio of $70.1 \pm 7.77\%$, compared to $61.0 \pm 4.38\%$ for the control strains. Importantly, strain G also outperformed the reference strain yREF, with a 1.47-fold increase in astaxanthin production and a 1.10-fold increase in astaxanthin selectivity. This indicates the merit of GEMbLeR compared to more conventional engineering approaches.

To correlate expression level changes with the underlying structural variation, we performed qPCR to quantify gene expression and used nanopore sequencing to determine the structure of recombined GEMs (Fig. 4g, h). The expression of *CrtI*, *CrtW* and *CrtZ* was found to correlate with carotenoid titers (Supplemental Fig. 8b, Supplementary Data 1, pairwise correlations). Specifically, upregulation of *CrtI* expression was linked to increased levels of lycopene, canthaxanthin and astaxanthin (Pearson correlation test with $p = 0.0126$, 0.0231 and 0.0187, respectively) and upregulation of *CrtZ* expression correlated with increased astaxanthin levels (Pearson correlation test with $p = 0.00216$). Downregulation of *CrtW* and *CrtZ* expression on the other hand correlated with increased levels of β-carotene, which is expected as the β-carotene ketolase (*CrtW*) and hydroxylase (*CrtZ*) are both responsible for converting β-carotene into canthaxanthin and zeaxanthin, respectively[84] (Pearson correlation test with $p = 1.75e-08$ and 0.0945, respectively). The data indeed showed that some of the best-performing variants (strain Z, X and G) increased the expression of *CrtI* and *CrtZ* after recombination. Notably, they also showed downregulation of other pathway genes, thereby supporting the idea that optimal pathway performance is not necessarily obtained by overexpressing all genes.

Sequencing further demonstrated that GEMbLeR only induced structural variation via Cre-LoxPsym recombination, with no SNPs or InDels detected. Interestingly, multiple samples with the same GEM layout for a certain gene (e.g., *CrtI* expression in clones B*, A* and K) showed different expression levels for that specific gene. This might be due to altered availability of TFs caused by altered GEM structures surrounding the other targeted loci. Importantly, among the 30 randomly selected clones, only three pairs were genetically identical: P & Q, B* & H and A* & S. This demonstrates that GEMbLeR allows to generate a large pool of genetically diverse clones in a short period of time. Clones with the same structural variation showed similar profiles

of carotenoid production and gene expression (statistical tests reported in Supplementary Data 1, carotenoids). Furthermore, we included one strain (E) with a white color in our selection and confirmed that this variant produced lower levels of carotenoids compared to the starting strain. This decrease in carotenoid production was most likely due to the strong reduction in expression of the first three genes in the pathway, caused by deletions of all or the strongest 5′ GEM-blocks.

Similar to the results obtained for the single reporter gene (*yECitrine*), we observed that the total number of deletions (347) outnumbered other recombination events; with 84 inversions, 7 translocations and 4 duplications. We also found that the frequency of recombination varied among the different constructs. For instance, 134 deletions occurred in the 5′ GEM array of *tHMG1*, whereas only four deletions were detected in the 5′ GEM of *CrtZ* (Fig. 4i). Three factors might explain the observed differences: the bias towards red-colored clones in our selection, the different genomic loci in which the constructs were introduced, and the efficiency of the LoxPsym variant. The LoxPsym variant used upstream of *CrtZ*[63] also reduced the capacity of GEMbLeR for expression diversification of a single gene (*yECitrine*, Supplemental Fig. 4d). Additionally, we tested the influence of the genomic context on expression diversification by inserting the same fluorescent construct at different genomic loci and confirmed that GEMbLeR efficiently induced expression variation at all sites. However, we did observe a reduction in fluorescence variation at the *CrtZ* insertion site (ChrXIII), indicating that the genomic locus can also influence the recombination process (Supplemental Fig. 9). Finally, our sequencing data confirmed that the orthogonal LoxPsym sites prevented GEM-blocks from different GEMs from interacting with each other.

## Discussion

By enabling the production of various metabolites, microbial cell factories can contribute to reducing our fossil fuel consumption and help pave the way towards a more sustainable economy. These metabolites are typically synthesized by complex metabolic pathways involving multiple enzymes that work together in an assembly line fashion to convert substrates into desired products. Fine-tuning the expression of individual pathway genes has proven important for optimizing production yields and limiting byproduct formation[21,24]. However, rational tuning of expression levels has proven challenging, often relying on trial-and-error approaches that are labor- and time-intensive. Here, we developed a strategy called GEMbLeR that overcomes these challenges. GEMbLeR allows for the generation of large libraries of strain variants with diversified expression levels of multiple genes without requiring prior knowledge. By creating this diverse pool of variants, GEMbLeR enables the identification of variants that show optimal production characteristics.

Current approaches for targeting and altering gene expression profiles often rely on combinatorial assembly techniques, such as COMPASS[44] and VEGAS[24], or high-throughput CRISPR-based approaches, such as CRISPR-AID[49] and BETTER[47], to generate a pool of strains with altered expression profiles. While these strategies have proven to be effective, their implementation is hampered by the design and cost of the required oligonucleotide libraries, as well as technical complexities. GEMbLeR avoids these drawbacks and simply requires a one-step pathway assembly reaction followed by a short Cre recombinase induction period to induce structural variation and generate a library of cells with highly diversified expression profiles for sets of target genes. Importantly, GEMbLeR does not rely on highly technical or costly equipment or reagents, making it easy to implement in a standard laboratory setting. Moreover, GEMbLeR can be used iteratively to further improve production traits of superior variants.

Our study showed that GEMbLeR allows to diversify expression over a wide range, spanning from complete shutdown of gene expression to levels higher than those obtained from the *TDH3* promoter, which is generally considered one of the strongest yeast promoters[71]. Covering expression diversification in the low levels is crucial for avoiding potential toxicity associated with certain metabolic steps[18,85]. Importantly, the range was covered in an almost continuous manner, allowing to obtain variants that only show subtle, but potentially important differences in gene expression. This continuity is achieved by combining 5' and 3' GEMs, as well as by using symmetrical LoxPsym variants, which ensure that the recombination outcome is independent of the LoxPsym orientation and allows to generate a much larger pool of diversified GEM-layouts compared to the canonical non-symmetrical LoxP site. We showed that LoxPsym-mediated inversions were crucial for obtaining this dense coverage, as we observed that inverted elements contributed to small expression modifications. In fact, our findings are consistent with observations from RNAseq experiments showing that eukaryotic promoters are often bidirectional[86,87], explaining why inverted elements still play a role.

To demonstrate the potential of GEMbLeR for strain optimization, we set out to improve astaxanthin production titers in *S. cerevisiae*. After only one round of GEMbLeR and testing a relatively limited number of variants, we identified several clones with a two-fold increase in production titers. Notably, our heterologous pathway included a bifunctional phytoene synthase/lycopene cyclase (*CrtYB*), that is needed for two non-consecutive pathway steps. To further improve pathway flux, it would probably be better to use two separate enzymes for these steps. This would allow their expression levels to be balanced individually with respect to up- and downstream enzymes.

GEMbLeR is a black-box approach that relies on chance rather than rationale to improve pathway expression profiles. One might argue that this strategy, along with most existing approaches, is therefore only applicable for optimizing phenotypes or production of molecules that can be easily measured. However, diverse fast high-throughput screening methods have recently been developed, such as biosensors[88,89], microfluidic based approaches[90,91] and high-throughput mass spectrometry[92,93], which enable rapid identification of optimal GEMbLeR variants, even when large variant pools need to be screened.

One important consideration when using GEMbLeR is the repetitive usage of the GEM construct for multiple heterologous genes. While we have shown that this does not cause genomic instability when targeting six heterologous genes simultaneously, it does limit the applicability of GEM, which relies on the available pool of sixteen orthogonal LoxPsym variants and can therefore target a maximum of eight genes simultaneously[63]. Although this is sufficient for many heterologous pathways, expanding GEMbLeR to target more genes can be achieved through the design of an array of heterologous genes with alternating directions. This would enable dual use of GEM for two genes simultaneously, which would allow to target twice as many genes with the same amount of GEM constructs. Moreover, if there is a need to reduce the number of GEM repeats, some of the building blocks could be replaced by other UPEs or terminators of similar strength, either from this study or from other high-throughput screens on hybrid promoters and terminators[27,94]. This flexibility provides opportunities for customization based on specific pathway requirements.

Importantly, while we only tested GEMbLeR in *S. cerevisiae*, the technique should be easily expandable to other organisms since we have previously demonstrated functionality of LoxPsym variants in various host species, including prokaryotes and plants[63]. Applying GEMbLeR in other hosts would simply require characterization of host-specific *cis*-regulatory elements and LoxPsym site activity. Another interesting and easy expansion of GEMbLeR would be to incorporate an array of LoxPsym-flanked alleles for each pathway gene. This would enable the generation of variants that express different alleles of specific genes. It has been shown that alleles derived from different organisms can drastically improve microbial production titers[11,39,95,96] and combining allele and GEM shuffling would allow to tackle this extra layer of metabolic engineering.

In conclusion, GEMbLeR offers an efficient strategy for multi-plexed gene expression diversification in vivo in yeast. By leveraging site-specific recombination and orthogonal LoxPsym sites, GEMbLeR diversifies the layout of a GEM thereby randomizing gene expression in a simple yet functional and effective way. A key advantage of our approach is that it is fast, inexpensive and low-tech, making it highly applicable for routine implementation in metabolic engineering approaches to improve the performance of microbial cell factories.

## Methods

### General methods

Supplementary Data 1, strains shows an overview of the strains used in this study. Supplementary Data 1, oligo's and Supplemental Table 4 list the DNA oligonucleotides and plasmids used in this study. Oligo's were obtained from Integrated DNA Technologies (IDT). Plasmids were purified using the QIAprep Spin Miniprep Kit (Qiagen). DNA amplification was done by PCR using SapphireAmp Fast PCR mix (Takara Bio) or GXL (Takara Bio) DNA polymerase. Cloning was done by restriction/ligation or using Gibson Assembly (NEBuilder HiFi DNA Assembly Master Mix). After cloning, the mixture was transformed to *E. coli* (*DSHα*, NEB) by heat shock, following the protocol provided by New England Biolabs. Long DNA constructs were synthesized by Qinglan Biotech, BGI (Supplementary Data 1, constructs).

### Strains and growth conditions

*E. coli* was used for transformation of cloning reactions using the lab strain DH5α (NEB) and cells were grown in selective Luria Bertani (LB) medium (10 g/L peptone, 10 g/L NaCl, 5 g/L yeast extract, 50 µg/mL carbinicilin) at 37 °C. *S. cerevisiae* strains were constructed from the lab strain *BY4741*, which is an *S288C*-derivative laboratory strain with genotype *MATa his3Δ1 leu2Δ0 met15Δ0 ura3Δ0*. Cells were grown in YPD (10 g/L yeast extract, 20 g/L peptone, 2 g/L glucose), Synthetic Complete (SC) medium (0.79 g/L SCM, 6.7 g/L YNB) or SC-Histidine medium (0.77 g/L SCM-His, 6.7 g/L YNB). Carbon sources (glucose, glycerol, raffinose and galactose) were added at 2 %.

### Design and construction of GEM via Gibson Assembly

The GEM consists of 5' and 3' recombinable gene expression modules (GEM-blocks) that consist of LoxPsym-flanked UPEs or terminators, respectively. The sequences of all promoters, UPEs and terminators tested in this research can be found in Supplementary Data 1, promoter and terminator sequences, respectively. To determine the TSS of promoters, we used the eukaryotic promoter database (EPD). The boundaries of UPEs were set at 500 bp upstream of the start codon and

the TATA box. The TATA box was found looking for the strong consensus sequence TATAWAW, with exception of the *RPL13B* promoter that had the weak TWTWWA consensus sequence. The region from the TATA box until the start codon was defined as the core promoter. Terminator sequences were obtained as previously defined in literature[27,97]. All UPEs and terminator sequences were flanked by LoxPsym sites, and different LoxPsym sites[63] were used for the up- and downstream part of the GEM, as well as between different GEMs regulating different genes. A list of all LoxPsym variants used in this research can be found in Supplementary Data 1, LoxPsym. The complete sequence of 5' and 3' GEM (with interchangeable LoxPsym site sequence 5'-ATAACTTCGTATATTATATAATATACGAAGTTAT-3') can be found in Supplementary Data 1, promoter and terminator sequences, respectively.

5' and 3' GEMs were constructed separately using Gibson Assembly. UPEs and terminator parts were amplified from *BY4741* genomic DNA and LoxPsym sites were added on to these parts via overhangs added to the primers (Supplementary Data 1, oligo's). Overhangs also added homology tails (20–40 bp) needed for the Gibson Assembly. The assembly reactions were performed using the NEBuilder HiFi DNA Assembly Mix and following the protocol provided by the manufacturer (New England Biolabs). A schematic overview of the cloning scheme is shown in Supplemental Table 3. Reactions were transformed to chemically competent *E. coli* cells (home-made) that were first thawed on ice for 30 min, after which 2 μL of the Gibson/Golden Gate reaction was mixed with 25 μL of competent cells in an ice-cold 1.5 mL Eppendorf tube. After 30 min incubation on ice, the reaction was heat shocked for 30 s at 42 °C and chilled on ice for 5 min. A volume of 300 μL LB medium was added, and the tube was incubated at 37 °C for 60 min in a shaking incubator. Finally, 100 μL of cells was plated on pre-warmed (37 °C) LB medium containing the appropriate antibiotics and incubated at 37 °C for ON growth. Cloning was verified using Sanger sequencing (Eurofins Genomics).

### *S. cerevisiae* genomic integrations

All *yECitrine* reporter constructs were integrated at the *CAN1* locus of *BY4741-mCherry*, unless specified otherwise. All heterologous genes needed for astaxanthin production were integrated at several loci indicated in Supplementary Data 1, genes. All genomic integrations were performed using Cas9 expressed from pVL382 (P1 in Supplemental Table 4, Addgene Plasmid #111436). sgRNA oligo's were annealed (5 min at 95 °C and slow cool down to 12 °C), phosphorylated (T4 polynucleotide kinase, NEB) and ligated (T4 DNA Ligase, NEB) into a BsmBI (NEB) digested and dephosphorylated (CIP, NEB) backbone[98]. Oligo's used for sgRNA cloning are shown in Supplementary Data 1, oligo's. For integration at *CAN1*, we used two sgRNAs simultaneously (OF/R 21, 22), for all other integrations one sgRNA was used (OF/R 23-28). Introduction of a Cas9-mediated double strand break was repaired via the native homology directed repair mechanism using a repair template with homology arms of 25–70 bp. Repair templates were amplified using the GXL Primestar polymerase (Takara Bio). Genomic integrations were verified by junction PCR using the SapphireAmp Fast PCR mix (Takara Bio) and a template prepared by boiling the clone in 50 μL NaOH (0.02 M) (99 °C, 10 min). *S. cerevisiae* transformation was performed by growing 3 mL ON culture in 2xYPD (20 g/L yeast extract, 40 g/L peptone, 4 g/L glucose). 1 mL was inoculated into 50 mL 2xYPD for 3 h in flasks, after which cells were harvested (centrifugation for 3 min, 3019 × g) and consecutively washed with 10 mL and 1 mL 0.1 M lithium acetate (LiOAc). Cells were resuspended in 100 μL 0.1 M LiOAc. PCR amplified donor DNA (50 μL) and/or plasmid DNA (200 ng) were added. A mixture containing 620 μL 50 % PEG 3350, 4 μL salmon sperm DNA and 90 μL 1 M LiOAc was added and mixed by vortexing. Incubation for 30 min at 30 °C, 18 × g, after which 100 μL DMSO was added and cells were heat shocked for 15 min at 42 °C. Cells were harvested by centrifugation (3 min, 3019 × g) and washed with 5 mM CaCl2. Cells

were incubated for a 3 h recovery period at 30 °C, 18 × g and finally plated on selective medium.

### Fluorescence assay and analysis

All fluorescent strains were derived from *BY4741-mCherry* (with the exceptions of strains used in Supplemental Fig. 7). *BY4741-mCherry* carried a genomically integrated expression cassette (at the *YRO2* locus) for constitutive expression of the fluorescent reporter *mCherry*[99], which was used for calculating normalized fluorescence levels. The effect of several genetic constructs on yECitrine fluorescence was tested using Flow cytometry (Attune NxT Flow Cytometer and Auto Sampler), using the BL1-A channel (excitation at 488 nm and emission at 574 nm with 20 nm bandwidth). Cultured yeast cells (cultured in 100 μL in 96 well plates in SC 2% glucose, unless medium is specified otherwise) were diluted in focusing fluid and measured with a flow rate of 200 μL/min. Cytometry data was gated based on the FSC-H to FSC-A map to select for single cells and also on YL2-A (excitation at 561 nm and emission at 610 nm with 20 nm bandwidth) to remove cells that did not express the *mCherry* control. Analysis and gating steps were done using the FlowJo software with (non-) fluorescent control strains as a reference (Supplemental Fig. 1d). Raw fluorescence data was exported from FlowJo using the plugin ViolinBox. Calculation of normalized fluorescence was done using Eqs. (1) and (2):

$$normalized\ fluorescence_{strainX} = normalization\ factor * \frac{BL1A_{strainX}}{YL2A_{strainX}} \tag{1}$$

with

$$normalization\ factor = \frac{\frac{BL1A_{yECitrine-control}}{YL2A_{yECitrine-control}}}{\frac{BL1A_{BY4741-mCherry}}{YL2A_{BY4741-mCherry}} * \frac{BL1A_{yECitrine-control,reference plate}}{YL2A_{yECitrine-control,reference plate}}} \tag{2}$$

BL1A and YL2A represent the median fluorescence measured from the sample well. Fluorescence from *BY4741-mCherry* and *yECitrine-control* strains was used to adjust the (non-) fluorescent baselines across experiments to compare data obtained from different batches. Note that, for some experiments (Fig. 2b, f and Supplemental Fig. 9) it is indicated that the normalized fluorescence was calculated differently depending on the experimental setup.

### RNA extraction and quantitative real-time PCR (qPCR)

ON cultures were grown in 1 mL YPD 2% (96 deep-well plates) and diluted to an $OD_{600\ nm}$ 0.05 in a volume of 1 mL. Samples were harvested in the exponential growth phase ($OD_{600\ nm}$ 0.4–0.6) for RNA extraction using the MasterPure Yeast RNA Purification Kit (LGC Biosearch Technologies). cDNA was obtained using the QuantiTect reverse transcription kit (Qiagen) and used as a substrate for qPCR. Each qPCR reaction was set up using the protocol provided by the manufacturer and a total volume of 5 μL. Three biological repeats were analyzed in technical duplicate for each sample using the StepOnePlus Real-Time PCR System (Applied Biosystem) and a thermal protocol as followed: 95 °C (10 min), 40 cycles of 95 °C (15 s), 61 °C (1 min). Primers were designed using the PrimerQuest Tool (IDT) and we used *TAF10*, *ALG9* and/or *TFC1* as reference genes[100]. $C_T$ values were determined using the StepOne Software v2.3 (Supplementary Data 1, expression). $\Delta C_T$ values for each target gene were calculated by subtracting the average $C_T$ value of the technical duplicates with the average $C_T$ value of the reference gene(s). $\Delta\Delta C_T$ values for each target gene shown for the astaxanthin producing strains were calculated by subtracting the average $\Delta C_T$ value of the reference strain ySTART (obtained from three biological repeats) with the $\Delta C_T$ value of the target gene.

## Fluorescence assay using different environments

Strains were inoculated in biological triplicate in 100 μL SC 2% glucose in 96 well plates for ON inoculation at 30 °C. Afterwards, cells were washed with demi water and diluted to $OD_{600\,nm}$ 0.05 in different media for another ON growth. Samples were again washed and diluted in their respective medium until they reached an $OD_{600\,nm}$ 0.2–0.4. The conditions which were used included three industrial media: (synthetic molasses, wine must and wort (16 °P)), all used for growth at 30 °C. Next, YPD 2% and SC 2% glucose medium were used to compare aerobic growth and growth in an anaerobic chamber, both at 30 °C. Synthetic complete (SC) medium was also used to asses growth at 30 °C using different carbon sources (2% glycerol, 2% galactose) and to test different temperatures (12 °C, 30 °C; 37 °C) using 2% glucose. Prior to fluorescence measurements with the flow cytometer (Attune NxT Flow Cytometer and Auto Sampler), samples were diluted in focusing fluid. Normalized fluorescence was calculated using Eq. (3) and did not use mCherry correction to avoid introducing a medium dependent effect due to differential *mCherry* expression.

$$normalized\ fluorescence_{strainX, mediumX} = \frac{BL1A_{strainX}}{BL1A_{BY4741-mCherry}} \quad (3)$$

## GEMbLeR induction and determination of recombination events

To induce recombination of GEM and diversify gene expression, cells were subjected to several growth and washing steps (Supplemental Fig. 4). For each experimental setup, two strains were tested: one with the control backbone (without Cre, pSH47-His-Vec) and one with the plasmid with the pGAL1-Cre expression cassette (pSH47-His-Cre) (P2 and P3, Supplemental Table 4). Single colonies of each strain were inoculated in SC-His 2% glucose for ON growth (100 μL, 96-well plates). Cells were washed (centrifugation for 3 min, 3000 × g) with SC-His 2% raffinose and diluted to a final $OD_{600\,nm}$ 0.05 and grown ON. Cells were washed and diluted in SC-His 2% raffinose 2% galactose to $OD_{600\,nm}$ for induction of *Cre* expression. Cells were induced for 6 h (*GEM-yECitrine* strains) or 2 h (*ySTART* strain), unless indicated otherwise. After induction, cells were washed and diluted with SC 2% glucose for ON recovery (dilution 1/20), after which cells were plated on YPD to obtain single GEMbLeR variants and/or used for flow cytometry analysis to obtain population level data.

Structural variation present in single GEMbLeR variants was annotated to obtain frequencies of deletion, inversion, translocation and duplication. Frequencies were obtained by counting the identity of the recombination event for each LoxPsym-flanked DNA element separately and dividing this by the total number of LoxPsym-flanked elements present in the cells.

## Sanger sequencing of GEMbLeR controlled *yECitrine* variants

After induction of GEMbLeR in fluorescent strains, random colonies were selected and fluorescence was measured using flow cytometry. A diversified set of 23 clones was selected for each tested GEM layout based on fluorescence data. The diversified GEM constructs were amplified using the GXL Primestar polymerase (TaKaRa) and the DNA template was obtained by boiling the clone in 50 μL NaOH (0.02 M) for 10 min at 99 °C. All amplicons were sent for Sanger sequencing (Eurofins Genomics) with the same oligonucleotides used for amplicon generation (OF/R 73-74, Supplementary Data 1, oligo's). Structural variation was identified via alignment of the individual UPE and terminator sequences (GEM-blocks) to the obtained GEMbLeR variant sequences using the SnapGene software.

## Design and construction of astaxanthin biosynthesis pathway

Six heterologous genes needed for astaxanthin production were introduced in *BY4741* at different genomic loci (Supplementary Data 1, genes). The first four genes of the pathway (*tHMG1, CrtE, CrtI* and *CrtYB*) were amplified from the pLM494 plasmid[55] (P4, Supplemental Table 4, Addgene #100539) and enabled β-carotene production. The sequences of the two last genes of the pathway (*CrtW* and *CrtZ*) were obtained from ref. [95] and amplified from synthesized DNA constructs[95] (Qinglan Biotech, BGI, Supplementary Data 1, constructs). ORFs were coupled to their regulatory promoter/5' GEM and terminator/3' GEM sequences via transformation of three separate amplicons with internal overlapping homology arms to construct the base strain ySTART. For construction of the reference strain yREF, Gibson assembly was used to fused promoters and terminators to the ORFs first. For synthesized constructs (*CrtW, CrtZ*), one amplicon was used for transformation of the native promoter/terminator regulated ORFs for the construction of strain yREF. Oligonucleotides OF/R39-72 were used for amplification of the repair amplicons (Supplementary Data 1, oligo's).

## Growth assay of astaxanthin production strains

To compare growth of *BY4741, ySTART* and *yREF*, four biological replicates of each strain were inoculated in dilution series for ON growth in YPD at 30 °C (150 μL). Samples with the same $OD_{600\,nm}$ were selected and grown again in dilution series for ON in YPD at 30 °C. Next, samples with the same $OD_{600\,nm}$ were selected and inoculated at $OD_{600\,nm}$ 0.01. Growth was tracked by $OD_{600\,nm}$ measurements every 15 min for 72 h at 30 °C using the Bioscreen C.

## Laboratory evolution to assess genomic stability in ySTART

To verify if repetitive usage of the GEM construct caused genomic instability, we used ySTART (biological quadruplicates) in a short-term evolution experiment, similar to previous methods[101]. We grew ON cultures in 5 mL YPD in test tubes and inoculated at $OD_{600\,nm}$ 0.05 at day 0. Strains were grown for 15 days and diluted (1/1000) to fresh medium at different time points (day 1, day 2, day 3, day 5, day 6, day 7, day 9, day 12, day 13). OD measurements were used to calculate the number of generations. After 100 generations, samples were plated and 8 randomly selected single clones were selected for amplicon analysis of the genomic loci with GEM controlled heterologous genes inserted. GXL Primestar polymerase was used for PCR with primers OF/ R81-92. Gel electrophoresis was used to analyze amplicon length by comparison to those of the base strain ySTART.

## Color analysis of carotenoid producing GEMbLeR variants

After induction of GEMbLeR in the base strain ySTART, cells were plated to YPD. Next, single colonies were randomly selected using the PIXL Precision Microbial Colony Picker (Singer Instruments) and transferred to rectangular YPD plates (96 well layout). The outside border of the plate layout was filled with the base strain (to minimize color differences due to position effects). Plates were incubated for ON growth at 30 °C and a new replica was made using the ROTOR+ (Singer Instruments). This replica was incubated at 30 °C for 48 h and stored at 4 °C for 96 h. Plates were next scanned and image analysis to obtain RGB values for each clone was performed using MATLAB. Reported RGB values were obtained by division with 65535. Clones with diversified GB values (indicative for red-colored phenotypes) were selected for further analysis.

## Extraction of carotenoids from yeast samples

Pre-growth of ON cultures was done using 96 deep well plates and 1 mL YPD. Carotenoids were extracted from shake-flask cultures (grown for 72 h at 30 °C in 50 mL YPD), which were inoculated at $OD_{600\,nm}$ 0.05. 1 mL of the culture was transferred to a 2 mL light-protected Eppendorf tube (Safe-Lock Tubes, amber, Eppendorf) and samples were kept on ice during the extraction protocol. Cells were first washed twice with 1 mL ice cold water (centrifugation for 3 min at 4 °C, 3000 × g). After careful removal of the supernatant, 400 μL of glass beads (acid-washed, 425–600 μm, Sigma Aldrich) were added to the sample tubes. Next, 500 μL of acetone (≥99.8%, AnalaR

NORMAPUR, VWR) was added and cells were disrupted by vortexing (using the FastPrep-24, MP Biomedicals for 60 s at maximum speed). Centrifugation (10 min at 4 °C, 16,000 × g) was used to pellet the cell lysate and carotenoids in the acetone phase were harvested by transfer of 300 or 200 μL (batch dependent) to a clean 2 mL light-protected Eppendorf tube. The remaining cell lysate was again washed for 2 additional extraction rounds by adding 500 μL fresh acetone and repeating the vortexing, centrifugation and harvesting steps. The carotenoid extract was stored at −20 °C until LC-MS analysis. Carotenoids were extracted in several batches and a reference sample (ySTART) was included to each batch to control for the batch effect.

### Sample preparation for LC-MS

After storage, samples were centrifuged (10 min at 4 °C, 16,000 × g) and 100 μL was transferred to another light-protected Eppendorf tube and used for sample preparation. Acetone was removed using a speedvac and carotenoids were resuspended in 100 μL acetonitrile:methanol;70:30 (v/v) by vortexing. The solution was next transferred to MS vials with glass inserts. Similar method was used for the standard solutions (1 or 2 mg/mL) of all reported compounds (astaxanthin (Sigma Aldrich), canthaxanthin (Supelco), β-carotene (Sigma Aldrich) and lycopene (Supelco)), which were also stored at −20 °C and mixed, evaporated and resuspended in acetonitrile:methanol;70:30 (v/v) at a final concentration of 3 μM for each compound. This standard was diluted in acetonitrile:methanol;70:30 (v/v) to prepare the other standards for a calibration curve based on 0, 0.1, 0.25, 0.5, 1 and 3 μM concentrations.

### LC-MS measurements and analysis

Samples were analyzed using a Vanquish LC System (Thermo Scientific) coupled by an electrospray ionization source (HESI-II Probe, Thermo Scientific) to a Q Exactive Orbitrap Focus mass spectrometer (Thermo Scientific). To equilibrate the instrument and column, 10 injections of mock sample containing only acetonitrile (Merck): methanol (VWR Chemicals); 70:30 (v/v) preceded the sample measurements. Next, 10 μL sample was injected onto a Acquity UPLC -HSS T3 column (1. 8 μm; 2.1 × 150 mm, Waters) and subjected to an LC gradient method adapted from ref. 102. Starting conditions with 85% solvent A (acetonitrile:methanol;70:30,v/v) and 15% solvent B (MilliQ water) at a flow rate of 0.25 mL/min were kept until 3.2 min. Next, a linear increase to 100% solvent A was obtained at 4.8 min. At 11.2 min, the flow rate was linearly increased to 0.3 mL/min at 12.8 min. At 21.6 min, conditions were adapted by a linear decrease of the flow to 0.25 mL/min and a linear decrease to 85% solvent A at 23.9 min. The column was equilibrated at these conditions until 29.2 min. The temperature of the column was kept constant at 32 °C. The mass spectrometer operated in full scan (range [70.0000–1050.0000]) and positive mode using a spray voltage of 3.5 kV, capillary temperature of 320 °C, sheath gas flow rate at 45, auxiliary gas at 0, sweep gas at 2. AGC target was set at 3.0e + 06 using a resolution of 70,000. Data collection was performed using the Xcalibur software (Thermo Scientific). Ions selected for each compound were the [M + H]+ ions for astaxanthin and canthaxanthin, and the [M•]+ cationic radical ion for β-carotene and lycopene. The data analyses were performed by integrating the peak areas (El-Maven – Polly - Elucidata). A quality control sample was obtained by mixing small aliquots of all samples and was analyzed every fifteenth sample to correct for signal drift by linear regression.

### Nanopore sequencing of astaxanthin biosynthesis strains

Six target loci of 30 GEMbLeR variants and 10 control strains were amplified using the GXL Primestar polymerase (TaKaRa) to sequence the (diversified) GEM layouts controlling astaxanthin pathway gene expression. The primers used for amplicon generation included F (5′- T

TTCTGTTGGTGCTGATATTGC-3′) and R (5′-ACTTGCCTGTCGCTCTA TCTTC-3′) tags needed for downstream processing (OF/R 75-80, Supplementary Data 1, oligo's). Amplicons were visualized using gel electrophoresis and sent for Nanopore sequencing at the Neuromics Support Facility, VIB-UAntwerp Center for Molecular Neurology.

In total 50 μL of pooled amplicons per sample (40 samples, 6 amplicons per sample) were purified using AMPure XP beads on Biomek FxP liquid handler (both Beckman Colter) in V: V ratio of 1: 0.8. After final elution, concentration was measured (Qubit) before barcoding PCR was performed on each pool using LongAmp Taq (NEB) and 40 barcodes from PCR Barcoding Expansion 1–96 EXP-PBC096 (ONT), as specified in ONT protocol. After barcoding and additional AMPure XP purification, the amplicons were visualized and quantified using Fragment Analyzer (using DNF-492 kit, both Agilent) and equimolarly pooled into a final pool that underwent the library prep. Library prep was based on updated protocol Genomic DNA by Ligation (SQK-LSK109) - Flongle Version: GDE_9063_v109_revT_14Aug2019 (ONT) and included FFPE repair during end prep. After library prep, 20 fmol of loading library was loaded onto the Flongle flowcell (FLO-FLG001). Sequencing was run of 23 h and generated 94.55 k reads (310.57 Mb), with N50 of 4.45 kb and on average 2233 reads per sample (372 reads per amplicon) and minimally 1000 reads per sample and 80 reads per amplicon (Supplementary Data 1, nanopore).

Basecalling of the Nanopore data was performed using the Guppy basecaller version 5.1.15 (Supplemental Fig. 10). Further analysis was performed using a pipeline integrated in genomecomb[103]. The sequences of all targeted GEMbLeR loci (with the six heterologous genes) of ySTART were combined into one fasta file (as "chromosomes") that was indexed and used as a reference for aligning all reads using minimap2[104]. The resulting sam file was sorted and converted to bam using samtools[105]. The number of reads mapping to each reference sequence was determined using a genomecomb query on the bam to find all unique combinations of chromosome and read name, followed by a query to count the number of readnames per chromosome. Structural variants were called using sniffles[106], cuteSV[107] and npinv[108]. SNV calls and haplotype separation of the bam were performed using longshot[109] and SNVs and small indel calling using clair3[110]. The resulting variant sets of different samples were combined and annotated using genomecomb[103]. Deviations from the reference genome which were present in all samples (including ten control samples) were not taken into account for further analysis. Structural variation calls were made when the frequency of variant reads was higher than 0.28. The minimal genotype quality (cuteSV) present in the data was and 21.5.

### Statistical analysis and data representation

Plots were constructed using the FlowJo software version 10.6.2, R package ggplot2[111] and Inkscape. All statistical analysis was done using R version 4.3.0. To determine normality of the data, we applied Shapiro Wilk's method (R package stats, function shapiro.test). To analyze statistical differences between multiple samples we first fitted a linear model (R package stats, function lm) and used analysis of variance (R package stats, function anova and R package emmeans[112], function emmeans) and two-sided post-hoc test Tukey multiple comparison of means (R package stats, function TukeyHSD). Post-hoc test for multiple comparison of means using the Dunnettx method was used for comparison between all samples and a control group (R package stats, function contrast, dunnettx adjustment). For pairwise comparison between the average fluorescence of two samples, we used an unpaired two-sided two sample t test (R package stats, function t test). For pairwise comparison between the variance of two samples, we used the non-parametric Fligner Killeen test (R package stats, function fligner.test). To construct correlation matrices we used function ggpairs (R package GGally[113]). To fit a linear regression we applied

function stat_poly_eq (R package qqpmisc[114]). To test the effect of environmental conditions on promoter and terminator strength, we constructed two linear mixed effect models (R package lme4[115], function lmer). Both models included the number of measured events as weight and the medium and promoter/terminator identity as random factors. In addition, one of the two models also included the interaction between the promoter/terminator and the medium as an extra random factor. These models were next compared using the function PBmodcomp (R package pbkrtest[116]) and function exactRLRT (R package RLRsim[117]) to test the significant difference of the interaction factor.

### Reporting summary

Further information on research design is available in the Nature Portfolio Reporting Summary linked to this article.

## Data availability

All data supporting the findings of this study are available within the paper and its supplementary information files. Source data are provided with this paper. Nanopore and Sanger Sequencing data have been deposited to NCBI Sequence Read Archive database under accession code PRJNA1000506. Source data are provided with this paper.

## Code availability

Code for analyses in this study (Guppy basecaller version 5.1.15) is provided in the Supplemental Information File.

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

## Acknowledgements

We thank Michiel Schreurs and Lloyd Cool for valuable feedback on statistical analysis. We also thank Dr. Karin Voordeckers for help with the manuscript. We also thank all Verstrepen lab members for valuable discussions. Thanks to the VIB Metabolomics Core Facility Leuven and the VIB Neuromics Support Facility Antwerp for help with experimental assays. C.C. acknowledges a PhD fellowship from FWO (1S25923N, 1SC2422N). J.Smets. was supported by VLAIO (HBC.2020.2623). A.Z. was supported PhD fellowship from Vlaams Instituut voor Biotechnologie (VIB). J.Steensels. acknowledges financial support from FWO by a postdoctoral fellowship (12W3918N, 12W3921N). D.D.R. and J.M. were supported by VIB and an FWO research grant (G061821N). A.G. was supported by a KU Leuven C1 grant (C16/17/006). Research in the lab of K.J.V. is supported by KU Leuven C1 Financing, VIB, VLAIO, FWO (G019223N) and iBOF (IBOF/21/092).

## Author contributions

C.C., J.Steensels, A.G. and K.J.V. conceptualized the study. C.C., J.M. and K.J.V. designed the experiments. C.C., J.Smets. and D.D.R. performed the experiments. C.C., P.B. and A.Z. contributed to the data analysis. C.C. and K.J.V wrote the manuscript and all co-authors contributed to review the manuscript.

## Competing interests

The authors declare no competing interests.
