## [Peer Review File · Nature Communications]

Reviewers' Comments:

Reviewer #1:

Remarks to the Author:

This paper describes a novel tool for in vivo gene expression diversification in yeast, called GEMbLeR (Gene Expression Modification by LoxPsym-Cre Recombination). With the orthogonal LoxPsym sites developed in their previous studies, they developed and characterized a pool of hybrid promoters and terminators flanked by LoxPsym sites, and used them to design a hyper-evolvable Gene Expression Modulator (GEM) construct at both 5' and 3' of the gene construct. They also demonstrate the GeMbLeR to generate a pool of yeast strains that can express fluorescent reporter with a diverse range of expression. Moreover, they applied the GEMbLeR in yeast cells to control astaxanthin biosynthesis pathway expression level, which result in a strain that can improve production titers more than two-fold. Although previous genetic engineering tools have been developed for similar purposes, this work is innovative in that it provides a tool that is cheap and simple to use to generate a pool of strains with a wide range of expression profiles. Thus, I do believe that the GeMbLeR could have a significant impact on both academic research and industrial manufacturing. Therefore, based on the outlined considerations, I support publication in Nature Communications after addressing the comments I have listed below.

Major comments

Limitations to yeast cells only: In the paper "Novel orthogonal LoxPsym sites allow multiplexed site-specific recombination in prokaryotic and eukaryotic hosts", it shows that the recombination efficiency is not correlated between different cell types. It is uncertain that how such a discrepancy will affect GeMbLeR applicability between eukaryotes and prokaryotes.

Minor comments

For the design of core promoter: line 195, combine UPEs of PGI1, TPI1, and TDH3 with the core promoter of the weak (PGI1) and strong (TDH3) promoter – better to have a figure to demonstrate the design.

For figure 2c: why are there two dots of TDH3, one is black, and one is purple?

Line 282: should it be non-recombine control or non-fluorescent control? It does not make sense that 41% GEM population have normalized fluorescence lower than the non-fluorescent control

Reviewer #2:

Remarks to the Author:

"Combinatorial optimization of gene expression through recombinase-mediated promoter and terminator shuffling in yeast" is part 2 to a companion paper titled "Novel orthogonal LoxPsym sites allow multiplexed site-specific recombination in prokaryotic and eukaryotic hosts" also submitted concurrently to Nature Communications by the same first and corresponding authors.

In this study Cautereels et al. report an approach for in vivo, multiplexed Gene Expression Modification by LoxPsym-Cre Recombination (GEMbLeR). GEMbLeR exploits orthogonal LoxPsym sites to independently shuffle promoter and terminator modules at distinct genomic loci. GEMbLeR is predicated on a system of reconfigurable promoter and terminator sequences (GEM-blocks), which can be assembled into arrays for the construction of a 5' and 3' GEM module. The 5' GEM consists of an array of upstream promoter elements separated by LoxPsym recombination sites, whereas the 3' GEM contains a set of different terminator sequences separated by LoxPsym sites. In turn, the authors demonstrate a modest proof-of-concept using the biosynthetic pathway of astaxanthin, an industrially relevant antioxidant, via a single round of GEMbLeR to achieve a moderate improved pathway flux and 2x production titers.

In general, the paper is mostly well written, and the data and figures are of high quality. I found the narrative to be interesting but a bit hard to follow, specifically with all of the rather cryptic references to the companion paper. Similar to my comment regarding the first paper, the narrative and results would benefit from a whole story (opposed to a set of companion papers).

In summary, Cautereels et al. titled "Combinatorial optimization of gene expression through

recombinase-mediated promoter and terminator shuffling in yeast" could be suitable for publication in Nature Communications. I recommend combining paper 1 with paper 2 and clearly articulating the role and incorporation of the "orthogonal LoxPsym sites" in the context of the metabolic engineering. I cannot recommend independent publication of the current work given the dramatic reduction in clarity and understanding of the context of the entire system when devoid of content from paper 1. Accordingly, I ONLY recommend publication of this paper upon the inclusion and thoughtful incorporation and assimilation of paper 1 (i.e., one paper clearly written and assimilated; NOT as a set of companion papers, which reduces the overall impact and clarity of both papers).

REVIEWER COMMENTS to paper NCOMMS-23-37042 (Combinatorial optimization of gene expression through recombinase-mediated promoter and terminator shuffling in yeast)

Reviewer #1 (Remarks to the Author):

This paper describes a novel tool for in vivo gene expression diversification in yeast, called GEMbLeR (Gene Expression Modification by LoxPsym-Cre Recombination). With the orthogonal LoxPsym sites developed in their previous studies, they developed and characterized a pool of hybrid promoters and terminators flanked by LoxPsym sites, and used them to design a hyper-evolvable Gene Expression Modulator (GEM) construct at both 5' and 3' of the gene construct. They also demonstrate the GeMbLeR to generate a pool of yeast strains that can express fluorescent reporter with a diverse range of expression. Moreover, they applied the GEMbLeR in yeast cells to control astaxanthin biosynthesis pathway expression level, which result in a strain that can improve production titers more than two-fold. Although previous genetic engineering tools have been developed for similar purposes, this work is innovative in that it provides a tool that is cheap and simple to use to generate a pool of strains with a wide range of expression profiles. Thus, I do believe that the GeMbLeR could have a significant impact on both academic research and industrial manufacturing. Therefore, based on the outlined considerations, I support publication in Nature Communications after addressing the comments I have listed below.

We would like to thank the reviewer for his/her constructive criticism and feedback indicating where the manuscript required improvement. Below, we provide a point-by-point response to the reviewer's comments and suggestions.

Major comments

Limitations to yeast cells only: In the paper "Novel orthogonal LoxPsym sites allow multiplexed site-specific recombination in prokaryotic and eukaryotic hosts", it shows that the recombination efficiency is not correlated between different cell types. It is uncertain that how such a discrepancy will affect GeMbLeR applicability between eukaryotes and prokaryotes.

We agree that applying GEMbLeR for combinatorial, multigene expression optimization in other species would be a great extension of the tool. In fact, we already hint at this in our discussion and mention that this would likely be feasible since the LoxPsym sites have been shown to work in other organisms. Applying GEMbLeR in another organism would indeed require the orthogonal LoxPsym sites to be tested in that specific organism of interest. Additionally, it also requires the development and assessment of new GEM-blocks, that comprise organism-specific *cis*-regulatory elements. Both of these DNA parts would together affect GEMbLeR performance in other hosts. We now added this comment to our discussion so that follow-up research can take this into account (lines 537-538). We consider the application of GEMbLeR in other organisms as a follow-up study that falls out of the scope of this research. The GEMbLeR system was developed as a tool to facilitate metabolic engineering of microbial cell factories, as it greatly accelerates the design and build steps of traditional laborious and costly DBTL-cycles. Microbes are increasingly used for sustainable production of various industrially relevant compounds and *S. cerevisiae* is one of the most widely used industrial microbes for such biotechnological applications^{1,2,3,4}. Therefore, we focused on *S. cerevisiae* for the development of our tool, similar to previous high-impact studies reporting tools for expression optimization in *S. cerevisiae*^{5,6,7,8}. We believe that the impact of our tool developed and tested for *S. cerevisiae* will be a

great addition to the field of synthetic biology and precision fermentation in *S. cerevisiae* and may inspire colleagues to expand the tool to other production organisms.

The revised text now reads (lines 535-538):

“Importantly, while we only tested GEMbLeR in *S. cerevisiae*, the technique should be easily expandable to other organisms since we have previously demonstrated functionality of LoxPsym variants in various host species, including prokaryotes and plants⁶³. Applying GEMbLeR in other hosts would require characterization of host-specific cis-regulatory elements and LoxPsym site activity.”

Minor comments

For the design of core promoter: line 195, combine UPEs of PGI1, TPI1, and TDH3 with the core promoter of the weak (PGI1) and strong (TDH3) promoter – better to have a figure to demonstrate the design.

We thank the reviewer for his/her suggestion. We have now added schematic representations of the hybrid promoter constructs to Supplemental Fig. 2 to clarify the designs. Additionally, we noticed that labels for UPE_{TPI1} and UPE_{TDH3} in panel b were switched and corrected this.

Revised Supplemental Fig. 2:

Supplemental Figure 2: Defining UPEs and core promoter for construction of the 5' GEM. **a.** Effect of promoter length - natural occurrence (purple) or limited to 500 bp (orange) - on the expression of *yECitrine*. Promoter sequences were tested in combination with the *CYC1* terminator and the fluorescent cassettes were integrated at the *CAN1* locus of *BY4741-mCherry*. The control (grey) represents strain *BY4741-mCherry* (no *yECitrine* cassette). Dots and bars represent average and standard deviation of the normalized fluorescence of three biological replicates. Statistics by two-sided Tukey multiple comparisons of means with p-values 0.9206, 0.008786 and 0.0000017 for *PGI1*, *TPI1* and *TDH3* promoters, respectively. **b.** Comparison of two core promoters (*PGI1* left and *TDH3* right) combined with their native upstream promoter element (purple), no UPE (black) or the LoxPsym-flanked UPE of *PGI1* (light purple), *TPI1* (blue) or *TDH3* (dark purple). Schematics at the right indicate the layout of the tested constructs, with a variable core promoter (either that of *PGI1* or *TDH3*). Statistics by Fligner-Killeen test with p-value 0.02547 to compare the variance of the samples using the *PGI1* core (variance 3.370) or the *TDH3* core (variance 1868). **c.** Correlation between the *yECitrine* fluorescence measured when expression was controlled by the native *S. cerevisiae* promoter (x-axis) or by the hybrid UPE_x-core_{TDH3} promoter (y-axis). Dots and bars represent average and standard deviation of the normalized fluorescence of three biological replicates. Linear regression with $R^2 = 0.29$ and p-value = 0.05838. **d.** Violin plots for comparison of the range in expression caused by hybrid promoters (i.e. LoxPsym flanked UPEs combined with the *TDH3* core promoter, left) or native promoters of *S. cerevisiae* (right). Dots represent the normalized fluorescence of one biological replicate and the color indicates the promoter sequence tested (three biological repeats per promoter sequence). Statistics by Fligner-Killeen test with p-value 0.7676 to compare the variance in fluorescence spanned

by the building blocks of GEM (variance 1054) or by our selection of native *S. cerevisiae* promoters (variance 1272). e. Final layout of the promoter part of GEM, consisting of six LoxPsym (yellow diamonds) flanked UPEs upstream of the *TDH3* core promoter. The numbers below each UPE indicates the length (bp) of the fragment. The sequences of UPEs can be found in **Supplemental Table 1**.

For figure 2c: why are there two dots of TDH3, one is black, and one is purple?

We thank the reviewer for pointing this out. The black dot indicates the native *TDH3* promoter and the purple dot indicates the hybrid promoter (i.e. a LoxPsym-flanked UPE_{TDH3}+core_{TDH3}). This was already indicated in the Figure legend, but we now adapted our wording and also added the information to the figure panel itself to clarify this.

The revised Fig. 2 c:

Figure 2: Selection of promoter and terminator GEM-blocks with diverse activity. c. Correlation between yECitrine fluorescence and mRNA abundance (ΔC_T value) of selected 5' GEM-blocks. Dots, horizontal and vertical error bars represent average, standard error and standard deviation of three biological replicates, respectively. Colors (shades of purple, blue and green) correspond to the UPE element used in the hybrid promoter with the *TDH3* core. Additionally, grey represents the *TDH3* core promoter by itself and black the native *TDH3* promoter.

Line 282: should it be non-recombine control or non-fluorescent control? It does not make sense that 41% GEM population have normalized fluorescence lower than the non-fluorescent control

We thank the reviewer for this comment. We understand the confusion, but indeed for 41% of the population we obtained normalized fluorescence levels that were lower than that of the non-fluorescent control, indicating that the cells lost yECitrine fluorescence. Values lower than those of the non-fluorescent control may be explained by noise. To avoid confusion, we revised our wording and changed it to “Surprisingly, 41 % of the diversified 5' GEM population did not show detectable yECitrine fluorescence (< the non-fluorescent control signal)” (lines 284-286).

Reviewer #2 (Remarks to the Author):

“Combinatorial optimization of gene expression through recombinase-mediated promoter and terminator shuffling in yeast” is part 2 to a companion paper titled “Novel orthogonal LoxPsym sites allow multiplexed site-specific recombination in prokaryotic and eukaryotic hosts” also submitted concurrently to Nature Communications by the same first and corresponding authors.

In this study Cautereels et al. report an approach for in vivo, multiplexed Gene Expression Modification by LoxPsym-Cre Recombination (GEMbLeR). GEMbLeR exploits orthogonal LoxPsym sites to independently shuffle promoter and terminator modules at distinct genomic loci. GEMbLeR is predicated on a system of reconfigurable promoter and terminator sequences (GEM-blocks), which can be assembled into arrays for the construction of a 5' and 3' GEM module. The 5' GEM consists of an array of upstream promoter elements separated by LoxPsym recombination sites, whereas the 3' GEM contains a set of different terminator sequences separated by LoxPsym sites. In turn, the authors demonstrate a modest proof-of-concept using the biosynthetic pathway of astaxanthin, an industrially relevant antioxidant, via a single round of GEMbLeR to achieve a moderate improved pathway flux and 2x production titers.

In general, the paper is mostly well written, and the data and figures are of high quality. I found the narrative to be interesting but a bit hard to follow, specifically with all of the rather cryptic references to the companion paper. Similar to my comment regarding the first paper, the narrative and results would benefit from a whole story (opposed to a set of companion papers).

In summary, Cautereels et al. titled “Combinatorial optimization of gene expression through recombinase-mediated promoter and terminator shuffling in yeast” could be suitable for publication in Nature Communications. I recommend combining paper 1 with paper 2 and clearly articulating the role and incorporation of the “orthogonal LoxPsym sites” in the context of the metabolic engineering. I cannot recommend independent publication of the current work given the dramatic reduction in clarity and understanding of the context of the entire system when devoid of content from paper 1. Accordingly, I ONLY recommend publication of this paper upon the inclusion and thoughtful incorporation and assimilation of paper 1 (i.e., one paper clearly written and assimilated; NOT as a set of companion papers, which reduces the overall impact and clarity of both papers).

We thank the reviewer for his/her thoughtful review of the paper and his/her kind words and acknowledgement of the high quality of the research. We specifically appreciate the reviewer for raising two points of criticism, being (1) a difficult narrative caused by an overload of references made to our other, companion study and (2) a reduced clarity and understanding of the context when devoid from this other study. Below and in the revised manuscript, we addressed the reviewer's comments.

We understand the reviewer's remark that the narrative might be a bit hard to follow due to many references to the companion paper. Therefore, we adapted the text and left out unnecessary/repeating references to the companion paper from the manuscript (specifically at lines 95, 122, 301, 436). Specifically, we mention the companion paper for four relatively simple points:

1. The number of genes which can be targeted using GEMbLeR depends on the number of available orthogonal recombination sites (lines 120, 524).
2. The cause of enhanced GEM deletion when testing 5' and 3' GEM separately is linked to the relative higher activity of outer recombination sites when positioned in an array (line 290).

3. The regulatory capacity of GEMbLeR reduces when using recombination sites that are less active (line 295).
4. Expanding GEMbLeR to other organisms could be feasible, as the recombination sites have been tested in other organisms as well (line 534).

The scope of our study on novel LoxPsym recombination sites goes significantly beyond the aspects listed above (e.g. defining determinants of recombination efficiency through computational modeling, deciphering new causes for recombination site orthogonality and testing recombination site orthogonality in different organisms, other than *S. cerevisiae* on which this paper focusses). As already outlined in our reply to comments raised on our companion paper, the message of the companion paper is completely different, focusing on the characterization of novel orthogonal recombination sites for genome editing in a wide range of organisms (in some cases allowing bypassing the use of Cas enzymes and the complexity of the corresponding IP). Therefore, we firmly believe that adding our other study on to this one will prevent us to clearly convey both distinct messages and highlight the full potential of each of the two systems. We feel that combining both studies into only one paper would reduce the clarity of both research lines, and, more importantly, reduce the impact of both papers. Moreover, we also want to point out that both papers are already quite lengthy, including several (supplemental) figures and tables.

References

1. Naseri, G. A roadmap to establish a comprehensive platform for sustainable manufacturing of natural products in yeast. *Nat Commun* **14**, 1916 (2023).
2. Naseri, G. & Koffas, M. A. G. Application of combinatorial optimization strategies in synthetic biology. *Nature Communications* vol. 11 Preprint at <https://doi.org/10.1038/s41467-020-16175-y> (2020).
3. Szymanski, E. & Calvert, J. Designing with living systems in the synthetic yeast project. *Nat Commun* **9**, 2950 (2018).
4. Cravens, A., Payne, J. & Smolke, C. D. Synthetic biology strategies for microbial biosynthesis of plant natural products. *Nat Commun* **10**, 2142 (2019).
5. Naseri, G., Behrend, J., Rieper, L. & Mueller-Roeber, B. COMPASS for rapid combinatorial optimization of biochemical pathways based on artificial transcription factors. *Nat Commun* **10**, 2615 (2019).
6. Liu, W. *et al.* Rapid pathway prototyping and engineering using in vitro and in vivo synthetic genome SCRaMbLE-in methods. *Nat Commun* **9**, (2018).
7. Barbieri, E. M., Muir, P., Akhuetie-Oni, B. O., Yellman, C. M. & Isaacs, F. J. Precise Editing at DNA Replication Forks Enables Multiplex Genome Engineering in Eukaryotes. *Cell* **171**, 1453-1467.e13 (2017).
8. Mitchell, L. A. *et al.* Versatile genetic assembly system (VEGAS) to assemble pathways for expression in *S. cerevisiae*. *Nucleic Acids Res* **43**, 6620–6630 (2015).

Reviewers' Comments:

Reviewer #1:

Remarks to the Author:

I found the response to my comments for this and the companion manuscripts satisfactory. They complement each other. Good job.

Reviewer #2:

Remarks to the Author:

The revised manuscript did not address my major concerns - i.e., combining paper one and paper two.

Accordingly, I will leave the final decision with the editor.